

# Uncertainty in Amazon vegetation productivity in CMIP6 projections driven by surface energy fluxes

Matteo Mastropierro[1], Daniele Peano[2], Davide Zanchettin[1]

[1] Department of Environmental Sciences, Statistics and informatics, Ca' Foscari University of Venice, Venice, Italy
[2] Fondazione Centro euro-Mediterraneo sui Cambiamenti Climatici, CMCC, Bologna, Italy

*Correspondence to*: Matteo Mastropierro (matteo.mastropierro@unive.it)

**Abstract.** The Amazon basin rainforest is a critical component of the climate system, currently representing 25% of terrestrial carbon gains and storing 150 to 200 billion tonnes of carbon. If and by which extent the Amazon rainforest will remain a net carbon sink is an open scientific question, motivated by the unexplained diversity across Earth System Model (ESM) results. Specifically, divergent responses are observed in Amazon vegetation productivity projections, especially under sustained global warming scenarios. We explore this inter-model diversity in projected Amazon vegetation in CMIP6 historical and ssp585 scenario simulations with thirteen ESM by explicitly accounting for the relative contributions of changes in the El Niño-Southern Oscillation (ENSO) and local mean-state climate changes. Our results demonstrate the dominant role of local mean-state climatic changes in shaping the response of the Amazon carbon cycle for 7 out of 13 ESM, with only a minor role for changes in ENSO and its teleconnection despite the strong inter-model diversity in representing ENSO. While temperature and water availability influence displays a high inter-model agreement, the most critical local processes determining uncertainty and divergence across ESM responses within the Amazon basin are the surface energy balance components, in particular shortwave incoming radiation and latent heat fluxes. We identify the main sources of model specificities in land scheme parameterizations, especially the incorporation of Phosphorous limitation, which leads to a stronger reduction of vegetation productivity under strong warming scenarios. We therefore advocate for increased focus from modelling groups towards a more accurate and consistent representation of surface radiative and turbulent fluxes in the Amazon region. Additionally, we hypothesize that a uniform incorporation of Phosphorous limitation across all the ESM may contribute to minimize the uncertainties. This dual approach can lead to more robust estimates of vegetation productivity within the Amazon basin across different climate change scenarios.



## 1 Introduction

The Amazon basin rainforest plays a fundamental role in the climate system, serving as a prominent actor in the land carbon cycle and by exerting a significative influence on the global energy budget and hydrological cycle (**Davidson *et al.*, 2012**). At the same time, the land carbon sink represents one of the crucial uncertainties affecting climate change future evolution (**Friedlingstein *et al.*, 2006; K. Arora *et al.*, 2020; Canadell *et al.*, 2021**). Indeed, projections of the Amazon climate and carbon sink are still poorly constrained by state-of-the-art Earth System Models (ESMs), indicative of persisting gaps of knowledge regarding this critical aspect of the Earth system (**Ahlström *et al.*, 2012; Zhu *et al.*, 2019; Xu *et al.*, 2020; Baker *et al.*, 2021; Koch, Hubau, *et al.*, 2021; Lin *et al.*, 2023; Raoult *et al.*, 2023**). In this paper, we characterize some of the key uncertainties affecting ESMs in the Amazon basin and identify the likely causes of inter-model discrepancies in the representation of Amazon vegetation productivity in a high-radiative forcing climate-change scenario.

The Amazon ecosystem has been a long-term carbon sink during the past decades contributing to approximately 25% of terrestrial carbon gains, estimated in 0.42-0.65 Gt C yr$^{-1}$ for the period 1990–2007, a trend mainly driven since the 1980s by the $CO_2$ fertilization effect from rising $CO_2$ concentrations in the atmosphere (**Phillips *et al.*, 2009; Pan *et al.*, 2011; O'Sullivan *et al.*, 2019; Walker *et al.*, 2021**). Nevertheless, recent estimates have demonstrated a slow-down of net carbon sequestration and consequently a saturation and declining trend in the Amazon carbon sink, with increase in carbon losses due to drought events and increased temperatures (**Brienen *et al.*, 2015; Hubau *et al.*, 2020**). Land carbon fluxes are commonly expressed in terms of Net Ecosystem Productivity (NEP), which is the result of productivity due to photosynthetic processes (GPP) minus Autotrophic (Ra) and Heterotrophic respirations (Rh). Both $CO_2$ concentrations in the atmosphere and climatic conditions affect land carbon fluxes. Higher atmospheric $CO_2$ concentrations exert mainly a positive direct effect on photosynthesis through plants stomatal closure and the associated negative Carbon-Concentration feedback (**Boer and Arora, 2009; K. Arora *et al.*, 2020**), while they can indirectly increase Ra and Rh (**Gao *et al.*, 2020**). Climatic conditions, on the other hand, mainly affect vegetation carbon fluxes through temperature and water availability, with a positive (negative) relationship between temperature (water availability) and changes in respiration rates (**Humphrey *et al.*, 2018; Gentine *et al.*, 2019; Green *et al.*, 2019; Liu *et al.*, 2020; Canadell *et al.*, 2021**). Additionally, the climatology of the Amazon forest influences the physical constraints on vegetation productivity: wetter parts of the forest (western and central Amazon) are primarily energy-limited, while those with a marked dry season (e.g., the Cerrado region) tend to be limited by water availability (**Huete *et al.*, 2006**). Interannual variations of water availability and temperature in the region are mainly related to El Niño-Southern Oscillation (ENSO), which is responsible for a vast part of the climatological and net land $CO_2$ observed interannual variability in tropical biomes (**Jones *et al.*, 2001; Kim *et al.*, 2016; Zhu *et al.*, 2017; Bastos *et al.*, 2018; Piao *et al.*, 2020; Mcphaden *et al.*, 2021a**). ENSO anomalies typically peak in boreal winter, coincident with the wettest portion of the year over the Amazon region (**Cai *et al.*, 2020**). Accordingly, some of the most severe droughts observed in the Amazon basin in recent decades and the associated reduction of the net land $CO_2$ sink were forced by strong warm ENSO events (or El Niños), most prominently including the 1997/1998 and 2015/2016 ones (**Jiménez-Muñoz *et al.*, 2016; Koren *et al.*, 2018;**





**Zhang *et al.*, 2019)**. Current explanations identify the increase of local temperatures forced by El Niño as the main process underlying the ENSO-Amazon connection, with potential consequences on the vegetation long-term state **(Jiménez-Muñoz *et al.*, 2016; Liu *et al.*, 2017; Bastos *et al.*, 2018; Zhang *et al.*, 2019)**.

Given these promises, at least three factors will contribute to determining whether, and to which extent, the Amazon ecosystem

will remain a net carbon sink in the future decades under sustained positive radiative forcing: mean-state climatic changes, nutrient limitation, and ENSO. First, a significant increase in surface air temperature and a marked decline in water availability in the Amazon basin as simulated for increased greenhouse gas emission scenarios will most likely result in a less effective carbon sink by the end of the 21st century **(Parsons, 2020)**. In particular, coupled climate models suggest that the reduction in precipitation is given by reduced evapotranspiration resulting from stomatal closure response to increased $CO_2$, which lead to

changes in local surface energy balance and atmospheric circulation patterns **(Kooperman *et al.*, 2018; Langenbrunner *et al.*, 2019)**. Then, Nitrogen and Phosphorous will likely more strongly limit tropical forests productivity **(Fleischer *et al.*, 2019)**, partly counterbalanced by the positive effect exerted by the increasing atmospheric $CO_2$ concentrations **(Huntingford *et al.*, 2013; Koch, Brierley, *et al.*, 2021)**. Lastly, an increased Amazon vegetation sensitivity to ENSO is expected under a range of global warming scenarios **(Kim *et al.*, 2017; Park *et al.*, 2020; Uribe *et al.*, 2023)**. In particular, projected changes in the

ENSO-Amazon connection may arise from both changes in ENSO properties (amplitude, skewness, spectrum) and from variations in the ENSO teleconnection mechanism with the Amazon region **(Chen *et al.*, 2017; Zheng *et al.*, 2017; Yeh *et al.*, 2018; Beobide-Arsuaga *et al.*, 2021; Cai *et al.*, 2021; Mcphaden *et al.*, 2021b)**. Notably, ENSO amplitude is slightly yet significantly enhanced under future global warming scenarios **(Beobide-Arsuaga *et al.*, 2021; Cai *et al.*, 2022)**, and regional patterns of precipitation and temperature anomalies over South America associated with ENSO teleconnections are projected

to be amplified in warmer climates **(Perry *et al.*, 2020; McGregor *et al.*, 2022)**. In this research we investigate the Amazon carbon sink in historical and ssp585 scenario conditions by means of CMIP6 ESMs simulations **(Eyring *et al.*, 2016; O'Neill *et al.*, 2016)** to separate the relative contributions of mean-state changes and changes in ENSO under sustained global warming. Specifically, throughout the paper we assess one question that remains underexplored in the literature: what are the relative contributions of ENSO and mean-state climatic changes to uncertainty in the projected Amazon carbon sink evolution? In

doing that, we additionally attempt to identify the local factors contributing to inter-model diversity in Amazon vegetation productivity.



## 2 Data and Methods

### 2.1 Data

We use simulations of the CMIP6 historical and ssp585 scenario experiments from thirteen ESMs (**Eyring** *et al.*, **2016**) (**Table 1**). We select only ESMs for which NEP values are available as model output or computable from other carbon fluxes. We use the first five realizations of each model, when more than one is available, with the caution of having the same simulation members for the historical and ssp585 scenarios to make a pairwise comparison. The details of the ESMs used are reported in Table SI1. All the analyses have been performed on single model realizations, and model means have been calculated solely

for the presentation of the results. The land carbon-cycle is investigated by considering monthly NEP values, which represent the balance of Gross Primary Productivity (GPP) due to photosynthesis at the net of autotrophic respiration (*ra*) and heterotrophic respiration (*rh*). Net Biome Productivity (NBP) is excluded due to possible inconsistencies in the representation of vegetation disturbance processes across different ESMs, such as Land Use Change (LUC) and fire dynamics. Overall, the following variables have been considered in the study: sea-surface temperature (*tos*), Net Ecosystem Productivity (*nep)*, Gross

Primary Production (*gpp)*, autotrophic respiration (*ra*), heterotrophic respiration (*rh*) , precipitation (*pr*), soil moisture (*mrso*), air surface temperature (*tas*), latent heat flux (*hfls*) and sensible heat flux (*rsds*).

**Table 1: Overview of ESMs used in the analysis**

| Institution | Model | Members (*hist*, *ssp585*) | Atmosphere and Land Resolution | Reference |
|---|---|---|---|---|
| IPSL | IPSL-CM6A-LR | 5, 5 | 1.27°N x 2.5°E | (**Boucher** *et al.*, **2020**) |
| CNRM-CERFACS | CNRM-ESM2-1 | 5, 5 | 1.4°N x 1.4°E | (**Séférian** *et al.*, **2019**) |
| AS-RCEC | CanESM5 | 5, 5 | 2.79°N x 2.81°E | (**Swart** *et al.*, **2019**) |
| MOHC | UKESM1-0-LL | 5, 5 | 1.25°N x 1.875°E | (**Sellar** *et al.*, **2019**) |
| MIROC | MIROC-ES2L | 5, 5 | 2.79°N x 2.81°E | (**Hajima** *et al.*, **2020**) |
| CSIRO | ACCESS-ESM1-5 | 5, 5 | 1.25°N x 1.875°E | (**Ziehn** *et al.*, **2020**) |
| BCC | BCC-CSM2-1 | 3, 1 | 1.1215°N x 1.125°E | (**Wu** *et al.*, **2019**) |
| E3SM-Project | E3SM-1-1-ECA | 1, 1 | 1°N x 1°E | (**Burrows** *et al.*, **2020**) |
| MPI-M | MPI-ESM1-2-LR | 5, 5 | 1.865°N x 1.875°E | (**Mauritsen** *et al.*, **2019**) |
| NCC | NorESM2-MM | 3, 1 | 0.94°N x 1.25°E | (**Seland** *et al.*, **2020**) |
| CCCma | TaiESM1 | 1, 1 | 0.94°N x 1.25°E | (**Wang** *et al.*, **2021**) |
| CMCC | CMCC-ESM2 | 1, 1 | 0.94°N x 1.25°E | (**Lovato** *et al.*, **2022**) |
| NCAR | CESM2-WACCM | 3, 5 | 0.94°N x 1.25°E | (**Danabasoglu** *et al.*, **2020**) |





Spatially-averaged values over the Amazon basin presented throughout the results are obtained by taking the spatial mean of the variable of interest within the Amazon basin shapefile downloaded from the SO HYBAM service **(INPE, 2019, https://hybam.obs-mip.fr/)**.

The ESMs performances in representing ENSO properties, the Amazon climatology, carbon and energy fluxes are evaluated against observational and quasi-observational products. The HadISST dataset is used for assessing ESMs sea surface

temperatures **(Rayner *et al.*, 2003)**, while ERA5 and ERA5-Land products are used to validate temperature, precipitation and soil moisture **(Hersbach *et al.*, 2020; Muñoz-Sabater *et al.*, 2021)**. Finally, the FLUXCOM-RS+METEO dataset, specifically the one forced with the WFDEI meteorological dataset **(Weedon *et al.*, 2014)**, is used as a reference for both carbon fluxes (GPP, NEP, Total Ecosystem Respiration, TER) and energy fluxes (shortwave incoming radiation and latent heat) **(Jung *et al.*, 2019, 2020)**.

To evaluate ESMs against the reanalysis products, a conservative remapping algorithm is applied to all the data to get a regular 1° longitude x 1° latitude grid, with the exception of the *tos* variable from ESMs with a curvilinear grid (Table SI1), for which a distance weighted (nearest-neighbor) average remapping is applied. The validation procedure refers to the climatological period 1979-2013. When comparing the carbon fluxes from ESMs with FLUXCOM data, the total ecosystem respiration is obtained by summing the contributions of *ra* and *rh*. An overview of the ESMs evaluation performances is available in the

supplementary material (Figures S1-S5).

## 2.2 Statistical analyses

### 2.2.1 ENSO index and composites

An annual time series of ENSO is obtained for each historical and ssp585 realization by averaging the corresponding monthly Nino3.4 index over the DJF season. The Nino3.4 index is defined as the 5-months moving average of spatially-averaged sea

surface temperatures over the region 170-120°W and 5°S-5°N, subsequently detrended by means of a 1st order polynomial and normalized. Years are then associated to one of the three main phases of ENSO, including the warm El Niño (EN), the cold La Niña (LN) and neutral conditions (N), for the historical period 1901-1960 (historical) and the future period 2041-2100 (ssp585). EN and LN events are identified by values exceeding the 90[th] and 10[th] percentiles of the DJF time series of the Nino3.4 index, respectively. Therefore, each of the EN and LN composites includes six events, for both historical and ssp585

scenarios. Values of the Nino3.4 time series within the 10[th] - 90[th] percentile range identify N years.

ENSO amplitude was simply defined as the standard deviation of the canonical Nino3.4 index.

### 2.2.2 Separation of mean-state and ENSO effects

To disentangle the relative contributions of ENSO changes and mean state (MS) changes, we adopt a framework conceptually similar to the one proposed by **(Power and Delage, 2018)**, that can be summarized with the following Eq. (1-3):






$$\Delta MS = N_{ssp} - N_{hist} \tag{1}$$

$$\Delta EN = \partial EN_{ssp} - \partial EN_{hist} \tag{2}$$

$$\Delta LN = \partial LN_{ssp} - \partial LN_{hist} \tag{3}$$

Therefore, mean-state changes ($\boldsymbol{\Delta MS}$) are defined as the difference between those years with neutral conditions in ssp585
($\boldsymbol{N_{ssp}}$) and the ones in the historical period ($\boldsymbol{N_{hist}}$). Similarly, $\boldsymbol{\Delta EN}$ represents the effects of El Niño under the ssp585 scenario with respect to the historical period, and results from the difference between El Niño-driven deviations from the ssp585 climatology ($\boldsymbol{\delta EN_{ssp}}$) and El Niño-driven deviations from the historical climatology ($\boldsymbol{\delta EN_{hist}}$). The same procedure was applied to estimate La Niña future impacts ($\Delta LN$). A Mann-Whitney U-test was used to evaluate the significance of $\boldsymbol{\Delta EN}$, $\boldsymbol{\Delta LN}$, and $\boldsymbol{\Delta MS}$.

**2.2.3 Effects of climatic drivers**

We additionally assess the relative contribution of ENSO and four climatic drivers (pr, tas, rsds, hfls) (eq 4) to the DJF interannual variability of NEP in spatially resolved data. To achieve this, we conduct two multivariate linear regressions (MLR): one in which the predictors and the dependent variable (NEP) are uniquely standardized (MLR-trend), whereas for the other all the data are linearly detrended and subsequently standardized (MLR-iav) Eq. (4). This allows us to estimate the
relative contributions of mean-state changes and interannual anomalies, respectively. However, we need to consider that a considerable part of NEP variability is controlled by ENSO, which in turn is highly correlated with the four local climatic drivers anomalies. Therefore, to discriminate the relative contribution of local climatic drivers with respect to ENSO modulation, we obtain the regressions predictors as residuals from the original variables after removing the linear contribution of ENSO, represented as the Nino3.4 index in Eq. (5-8). We apply a 5-fold cross-validation ridge regression model, because
the penalty score in the cost function of ridge regression helps to account for the collinearity among the predictors themselves. We chose the best performing regularization parameter among a set of values spaced evenly on a log scale. The analyses are performed using the Scikit-Learn package available for the Python programming language (**Pedregosa et al., 2012**).

$$NEP = a_0 + \beta_1\,\varepsilon_{pr} + \beta_2\,\varepsilon_{tas} + \beta_3\,\varepsilon_{rsds} + \beta_4\,\varepsilon_{hfls} + \beta_5 nino_{3.4} + \varepsilon \tag{4}$$

$$\varepsilon_{pr} = pr - a_0 - \alpha\,nino_{3.4} \tag{5}$$

$$\varepsilon_{tas} = tas - a_0 - \alpha\,nino_{3.4} \tag{6}$$

$$\varepsilon_{rsds} = rsds - a_0 - \alpha\,nino_{3.4} \tag{7}$$

$$\varepsilon_{hfls} = hfls - a_0 - \alpha\,nino_{3.4} \tag{8}$$





We applied a Mann-Whitney U-test of independence with Bonferroni correction to assess whether the zonal means of the regression coefficients within the Amazon basin are significantly different between historical and ssp585. To mitigate the risk of overstating the significance of the statistical tests conducted, we employ a false discovery rate (FDR) control method based on **(Wilks, 2016)**. This approach effectively addresses the issue of multiple hypothesis testing, ensuring a more accurate interpretation of the obtained results.

All the results reported in the manuscript are uniquely for the DJF season.

# 3 Results

## 3.1 Intermodel uncertainties of NEP and climatic drivers

Amazon basin vegetation productivity shows substantial overlap across models during the historical period but strongly diverging trends across models during the ssp585 scenario, with individual models differing in magnitude and sign of projected

changes (Figure 1a). The multi-model ensemble yields a mean of NEP by the end of the 21$^{st}$ century of 24.5 gCm$^{-2}$ and an inter-model standard deviation of 68.8 gCm$^{-2}$. Inter-model uncertainty is much higher than intra-model uncertainty, originated by ESMs internal climate variability and represented by the ±1 standard deviation spread of the projections in Figure 1. On the opposite, the climatological drivers of NEP present a stronger agreement among ESMs. A multi-model mean reduction of -24.6 mm month$^{-1}$, -132.7 kgm$^{-2}$ and -10.26 Wm$^{-2}$ is projected for precipitation, soil moisture and latent heat, respectively

(Figure 1b,c,f), while an increase is observed for temperature and incoming shortwave radiation (+7.4 °C and +10.1 Wm$^{-2}$, respectively, Figure 1d,e). The multi-model ensemble spread at the end of the 21$^{st}$ century remains substantial for all these variables. Differences among models of one or even two orders of magnitude could be seen for instance between CESM2-WACCM and CanESM5 for shortwave incoming radiation or for MIROC-ES2L and TaiESM1 regarding soil moisture projections. Overall, the signs of the projected changes in these variables are more coherent than for NEP, with the notable

exceptions of CESM2-WACCM for shortwave incoming radiation and CMCC-ESM2 for latent heat.

For some models, divergent NEP projections cannot be easily attributed to similar deviations projected for the individual climatic drivers. For instance, MIROC-ES2L and CanESM5 projects similar end-of-the-century NEP values, but strongly diverging trends of all other variables. On the other hand, MIROC-ES2L projects an extreme reduction in soil moisture against negligible changes in precipitation and a rather weak warming, yielding the strongest NEP increase. Lastly, despite the extreme

warming and reductions in precipitation and latent heat, CanESM5 yields a NEP increase similar to the multi-model mean.

Regarding the intra-model spread, the highest variability is observed in NEP, followed by precipitation and latent heat. Some models display a very limited variability in all the considered variables (e.g., CESM2-WACCM), whereas others (e.g., ACCESS-ESM1-5) show an extremely large spread in NEP projections and a moderate one in precipitation and shortwave incoming radiation. It is still unclear how the inter-model spread in NEP is directly related to the spread in individual climatic

drivers. All the carbon fluxes from which NEP is derived (GPP, Ra and Rh) depict an increasing trend throughout the 21$^{st}$ century (Figure S6). Among those carbon fluxes, GPP presents the highest inter-model standard-deviation (578.7 gCm$^{-2}$),



followed by Ra and Rh (371.47 and 225.9 gCm$^{-2}$ respectively). This shows that uncertainty in NEP does not solely stem from photosynthesis or respiration; instead, it arises from inconsistencies and limitations in how models represent both processes. Conversely, the intra-model uncertainty for GPP, Ra, and Rh exhibits a relatively low standard deviation, implying that the

significant intra-model variability observed in NEP in Figure 1a is shaped by the cumulative effect of individual factors. Lastly, a key point is how much the spread toward the end of the 21$^{st}$ century reflects internal interannual-to-decadal variability, and changes of it in a warmer climate, rather than differences in trends.

Overall, inconsistencies in projected Amazon NEP cannot be simply understood as a consequence of discrepancies in the projected vegetation climatic factors, both regarding trends in the mean values and intra-model uncertainties.


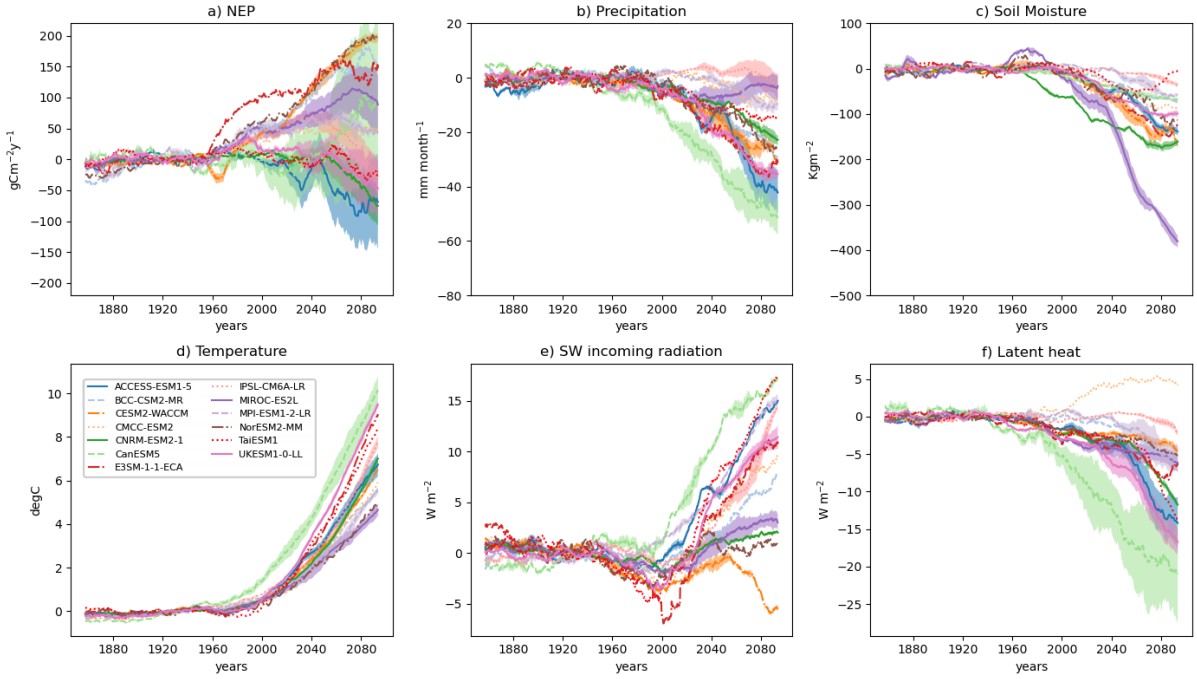

**Figure 1:** Simulated anomalies of (a) NEP, (b) precipitation, (c) soil moisture, (d) temperature, (e) SW incoming radiation and (f) latent heat in the Amazon basin for the *hist* and *ssp585* experiments. Anomalies are computed with respect to the 1901-1960 mean. For the models with more than one realization, both the model-ensemble mean (line) and the spread (±1 standard deviation, shading) are shown. 4 210 years moving average values are shown for clarity.

## 3.2 ENSO properties change

Changes in key properties of ENSO could significantly impact the Amazon basin region. Differences in the ENSO amplitude, represented by the Nino3.4 index standard deviation, between the ssp585 scenario (blue dots) and the historical period (red




squares) are shown in Figure 2. Nine out of thirteen ESMs show an increased ENSO variability under the ssp585 scenario, whereas CMCC-ESM2, TaiESM1 and UKESM1-0-LL do not project relevant changes, and BCC-CSM2-MR predicts a decrease in ENSO amplitude.


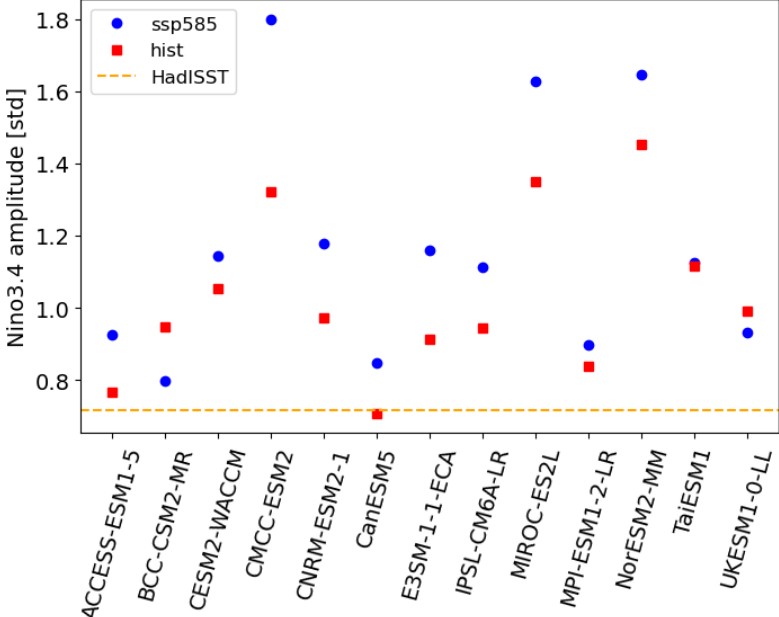

**Figure 2:** ENSO amplitude change as represented by the Nino3.4 signal standard deviation from the historical period (red squares) to the future ssp585 scenario (blue dots). The orange dashed line represents the value of Nino3.4 signal amplitude calculated for the HadISST dataset.


Several studies depict a possible increase in the frequency of extreme ENSO events under global warming (**Cai** *et al.*, **2014, 2015; Berner** *et al.*, **2020; Brown** *et al.*, **2020; Fredriksen** *et al.*, **2020**), which could have important implications on the Amazon carbon sink due to a stronger inhibition of the tropical teleconnection pathway. The ESMs in our ensemble largely overestimate the observed ENSO amplitude in the interannual-to-decadal band, with the associated spectra typically featuring

a narrow peak around the 3-year periodicity (Figure S7). The ESMs also yield a diversity of changes in ENSO spectral characteristics under the warming scenario: generally all the models represents a shift of ENSO signal toward higher frequencies, with weaker amplitude at decadal time scales and stronger amplitude at interannual time scales (Figure S7).

### 3.3 Impact of mean-state changes on NEP

Variations of NEP under warming scenario with respect to the historical period can be attributable either to mean state changes in local climatic conditions or to an increased impact of ENSO on vegetation productivity (**Kim** *et al.*, **2017,** see Methods).



Below, we show the impact on NEP of $\Delta MS$ in Figure 3, while El Nino ($\Delta EN$) and La Nina ($\Delta LN$) contributions are shown in Figure S8 to Figure S11.

Overall, most of the ESMs project an increase of NEP attributable to mean-state climatic changes in the Amazon region (Figure 3 and Figure 4), apart from ACCESS-ESM1-5 and UKESM1-0-LL, that show a decrease in vegetation productivity. In ACCESS-ESM1-5, the decrease is likely explained by the implementation of phosphorous limitation in its CABLE land surface model (**Ziehn *et al.*, 2020**), in addition to the nitrogen one, which is demonstrated to be a strong limiting factor on tropical vegetation productivity in the Amazon basin (**Terrer *et al.*, 2019; Davies-Barnard *et al.*, 2020; Ziehn *et al.*, 2020; Braghiere *et al.*, 2022**). Additionally, a NEP increase is observed in the Cerrado savannah ecosystem for all but three ESMs (ACCESS-ESM1-5, CanESM5 and CNRM-ESM2-1). Differences and similarities of responses across ESMs could partly be attributable to common land model characteristics: CESM2-WACCM and NorESM2, for example, rely on the same land module (CLM5), (Table SI1), and this explains the similar patterns of vegetation response in those two ESMs.

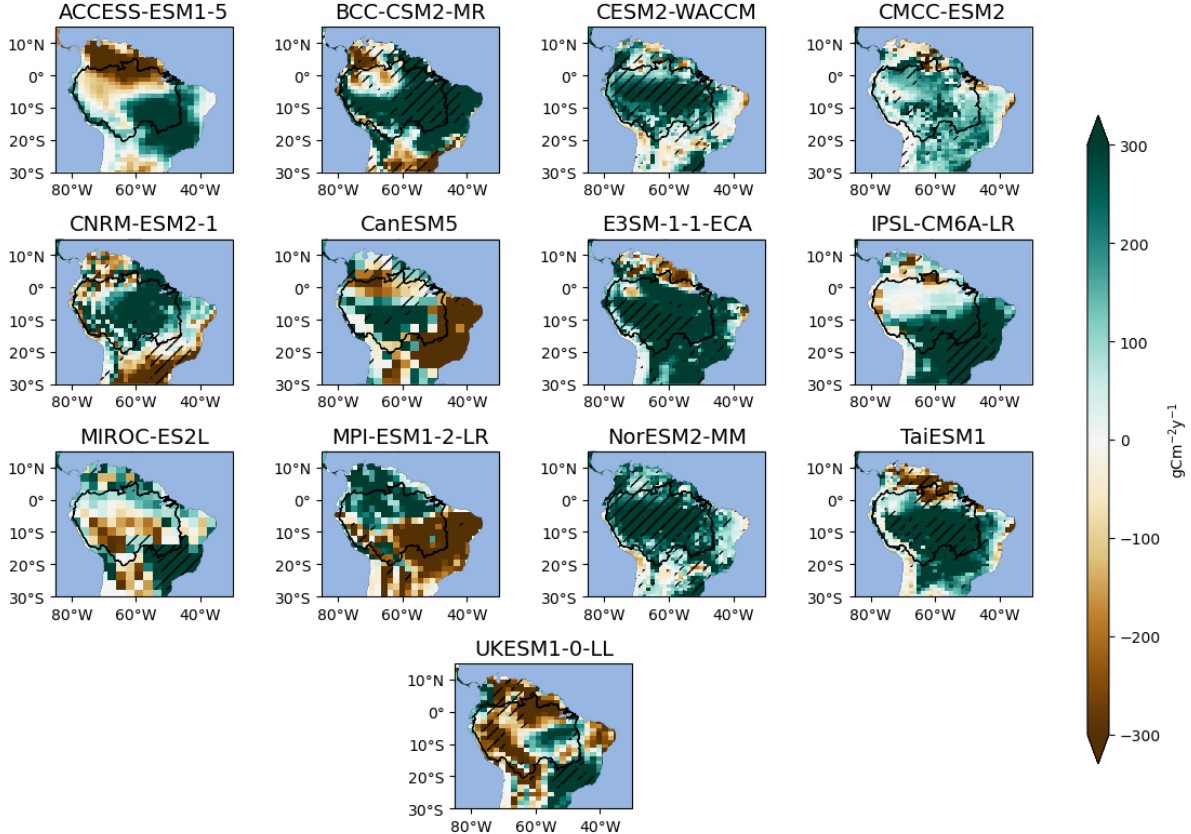

**Figure 3:** Mean-State Change in the *ssp585* scenario compared to the *hist* simulation ($\Delta$MS). Hatches indicate grid-cells for which the future mean state of NEP (Nssp) is statistically different from the historical NEP mean state (Nhist), according to a Mann-Whitney U-test. The Amazon basin, obtained from the SO HYBAM service (**https://hybam.obs-mip.fr/**), is also represented.



Overall, $\Delta MS$ impacts, attributable to both mean-state climatological changes and $CO_2$ fertilization effect on vegetation
productivity, are expected to dominate the vegetation productivity response in the ssp585 future scenario (Figure 4a). On the
other hand, despite a widespread model agreement on projected changes in the ENSO amplitude signal with consequent
stronger El Niño events (Figure 2 and **Cai *et al.*, 2014**), the effects on vegetation productivity of both the positive and negative
phases of ENSO are of comparable or higher magnitude with respect to mean-state changes for roughly half of the models,
namely ACCESS-ESM1-5, CMCC-ESM2, CNRM-ESM2-1, CanESM5, MPI-ESM1-2-LR and UKESM1-0-LL. Still, all the
ESM except for CMCC-ESM2 do project stronger negative anomalies associated to the El Niño phase in the ssp585 scenario,
although with lower magnitude and spatial extent with respect to $\Delta MS$. Exceptions are ACCESS-ESM1-5, CNRM-ESM2-1
and UKESM1-0-LL, for which $\Delta EN$ impacts are larger both in magnitude and in space compared to $\Delta MS$. While BCC-CSM2-
MR, E3SM-1-1-ECA and TaiESM1 depict a more spatially heterogeneous $\Delta EN$ impact, in IPSL-CM6A-LR, CNRM-ESM2-
1 and MIROC-ES2L the north-eastern amazon basin is the most impacted region, opposite to what is observed in ACCESS-
ESM1-5. Composites of La Niña effect ($\Delta LN$, Figure S11) depict an almost exact specular impact compared to El Nino effect,
with positive NEP anomalies within the Amazon basin especially for ACCESS-ESM1-5, CNRM-ESM2-1, CanESM5 and
UKESM1-0-LL, while E3SM-1-1-ECA, TaiESM1 and partly CESM2-WACCM depict overall an heterogeneous signal.

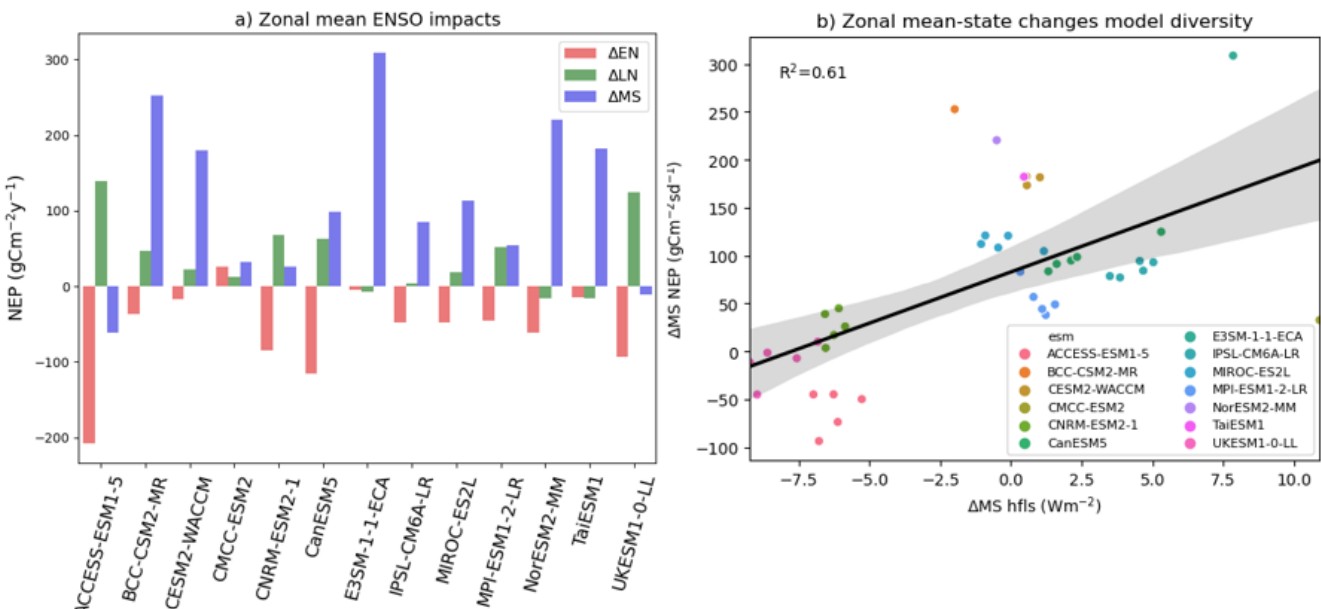

**Figure 4:** a) Spatially mean values of ΔMS (blue), ΔEN (red) and ΔLN (green) of NEP in the Amazon basin for individual ESMs; b):
Model differences in ΔMS of NEP as explained by ΔMS of latent heat flux.





Mean-state changes of NEP within the Amazon basin may be linked to different underlying climatic drivers. We therefore
compared the mean-state changes of NEP with the ones related to the considered predictors to both explain ΔMS impacts and
to discriminate different model behaviour. Figure 4b reports the linear relationships between the mean-state changes of NEP
and mean-state changes of latent heat within the Amazon basin, for each model simulation. It emerges that models that project
higher hfls typically tend to produce also an increase in NEP values by the end of the century.

### 3.4 Drivers of NEP


We explore the NEP response in the Amazon basin to long-term changes and interannual anomalies of four climatic drivers,
following the approach described in the Methods section. Overall, we found the performance of our regression models to be
satisfactory, as demonstrated by the fact that the interannual Amazon zonal NEP values in both the historical and ssp585
experiments are skilfully predicted for all the ESMs (see Figure S12 and Figure S13 for model performances).


### 3.4.1 Long-term changes effects on NEP

Figure 5 illustrates the regression coefficients values (from the MLR_trend regression) of the five climate drivers, averaged
for every model and within the Amazon basin, representing the long-term changes influence of the predictors on NEP. Long-
term changes of shortwave incoming radiation emerge as the most influential factor in driving a positive response of NEP for
all the ESMs, followed by a lower (positive) contribution of temperature. Despite the overall agreement in the sign of these
two climatic drivers, we observe a high intermodel variability, especially for *rsds*. CMCC-ESM2 and MIROC-ES2L, in
particular, have the lowest contribution of *rsds*, while E3SM-1-1-ECA, CanESM5, MPI-ESM1-2-LR display the highest
coefficients for *rsds*, and at the same time the lowest ones for *tas*. On the opposite, negative coefficients are observed, with a
few exceptions, for precipitation and ENSO. Again, very high variability emerges between the models, with MIROC-ES2L,
MPI-ESM1-2-LR and UKESM1-0-LL showing a slightly positive contribution from the ENSO signal contrary to all the other
ESMs. Most likely, the positive long-term temperature effect shown with different magnitude by the models, is influenced by
atmospheric $CO_2$ concentrations, which strongly correlates with temperature variations. Similarly, NEP sensitivity to
shortwave incoming radiation reflects its increasing trend resulting from a projected reduction in cloud cover, and consequently
precipitation, which we generally find to have a negative coefficient in Figure 5, within the Amazon basin (see also Figure
1e). Likewise, the negative long-term effect of precipitation can be attributed to its diminishing trend over the Amazon basin,
while the rising trend in tropical ocean temperatures that characterizes the future ENSO signal is strongly influenced by its
InterAnnual Variability (IAV), determining overall a negative effect. Lastly, contrasting results are found for latent heat, with
some models displaying a positive contribution (e.g., BCC-CSM2-MR and E3SM1-1-ECA), whereas others (e.g., ACCESS-
ESM1-5) showing a negative one, even though its coefficient values are generally lower in absolute terms compared to the
ones of the other predictors.

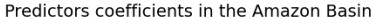

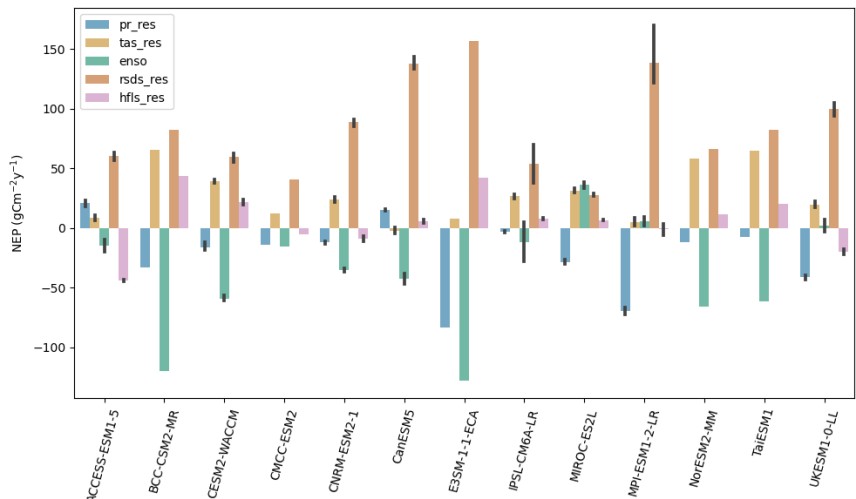

**Figure 5:** Regression coefficients values of climatic drivers averaged within the Amazon basin, from the MLR-trend model built by merging the data of the historical and ssp585 scenario. The coefficients here shown represent the long-term mean-state changes effects of the predictors on NEP. All the data have been standardized before the analysis. The black vertical bars represent the spread in the predictors' coefficients for models with more than one realization available.

### 3.4.1 Interannual anomalies effects on NEP

Figure 6 illustrates the distribution of models simulations for the Amazon basin spatially averaged values of NEP as predicted by the MLR_iav regression with respect to the regression coefficients of *pr* and *rsds*, for the historical period (see Figure S14 for the warming scenario *ssp585*). Precipitation, and more robustly shortwave incoming radiation, emerge as the most influential factors, at the interannual time-scale, in discriminating the behaviour of NEP within the Amazon basin among the ESMs. The positive relationship between the regression coefficients value of precipitation and the NEP predicted by the regression model across ESMs intuitively indicates that models with stronger (negative) precipitation signal generally predict an overall lower vegetation NEP. In contrast, a negative association is found between NEP and the effect of shortwave incoming radiation: ESMs featuring a net increase in NEP are typically less affected by shortwave incoming radiation. We therefore demonstrate that these two climatic drivers are the ones controlling for the NEP variability across models at interannual time-scales, and thus are vastly responsible for the observed intermodel uncertainty in modelled NEP.




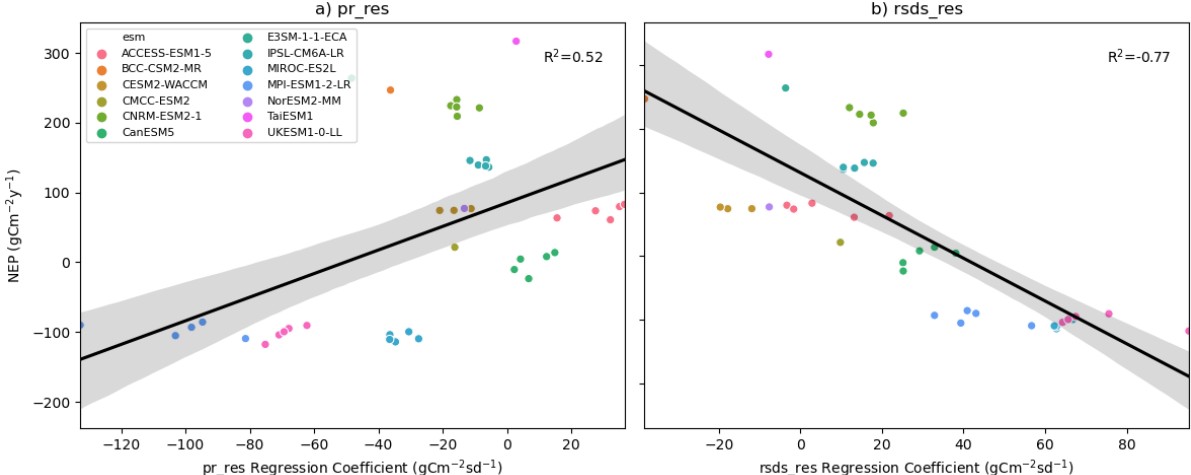

**Figure 6:** Model diversity in the representation of Amazon basin spatially averaged values for predicted NEP (on the y-axis), with respect to the regression coefficients of a) precipitation and b) shortwave incoming radiation, obtained with the MLR_iav regression. Shown are the values referring to the historical experiment. All the data have been detrended and standardized before the analysis.


The ESMs depict a diverse range of NEP sensitivities to the predictors considered, as can be seen from Figure S15 As expected, all the models show a negative influence of ENSO, temperature, and precipitation (for this, ACCESS-ESM-1-5 is the only one to display a consistent positive influence of precipitation) in driving NEP response at interannual time-scales, while on the opposite, a positive NEP sensitivity to *hfls* is consistently produced by all the ESMs. NEP sensitivity to rsds is found to be

positive especially for those models that, on average, project a weaker productivity within the Amazon basin, namely MIROC-ES2L, MPI-ESM1-2-LR and UKESM1-0-LL (Figure 7b, Figure S15).

Finally, we evaluate the spatially resolved regression coefficients (Figure S15) to identify regions of inter-model agreement and of model specificities. To do so, we re-gridded all the ESMs coefficients to a common 1x1 grid using bilinear interpolation.

The multi-model means of the *ssp585* scenario are reported in Figure 7. Hatches regions indicate where at least 10 out of 13 ESMs agree in the sign of the predictor value. As clearly discernible from Figure 7, *pr*, *tas* and ENSO are negatively correlated with NEP, while *hfls* and, to a lesser extent, *rsds* display a positive correlation. In particular, ESMs agree in the sign of the predictor's sensitivity over the vast majority of the Amazon basin in the case of *tas*, ENSO and *hfls*. Furthermore, a significant hotspot can be observed in the northern and northeastern regions of the Amazon basin, with strong inter-model agreement and

a tight relationship observed between NEP and ENSO, *hfls* and to a lesser extent *pr*.



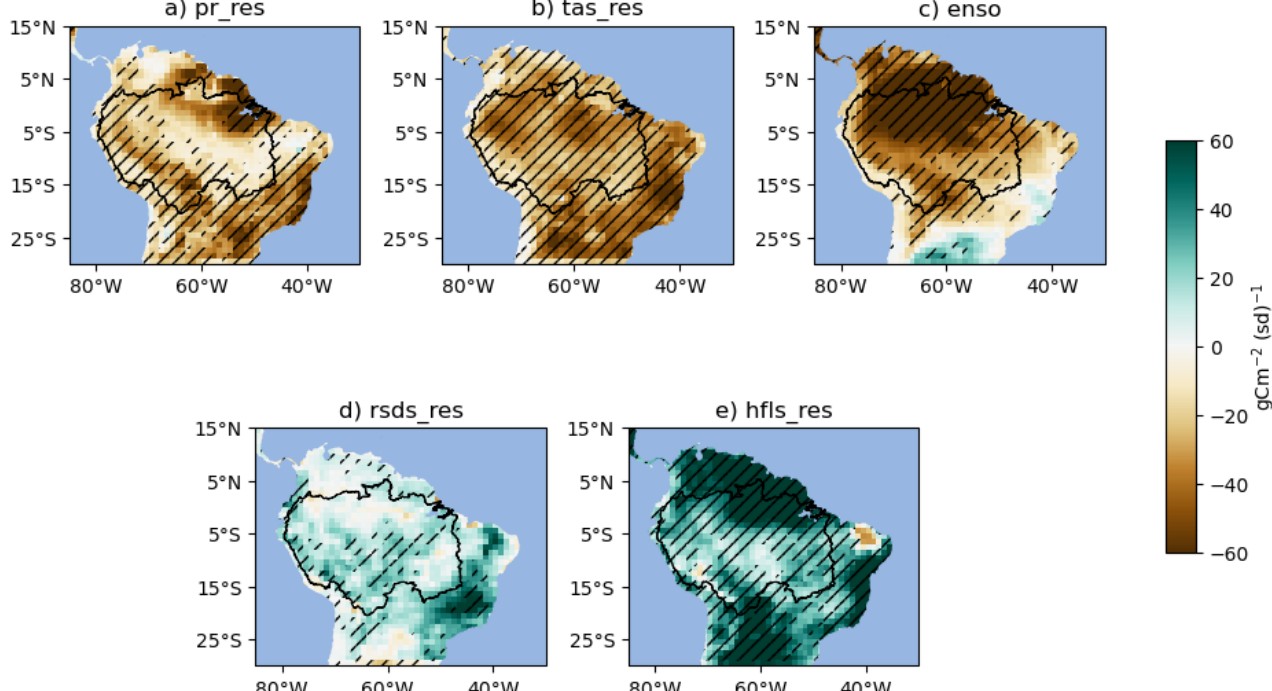

**Figure 7:** Multi model ensemble mean of the coefficient values for the five climatic drivers obtained by the MLR_iav regression, for the ssp585 period. Hatches represent those grid cells for which at least 10 out of 13 ESMs agree in the sign of the predictor value. The Amazon basin, obtained from the SO HYBAM service (**https://hybam.obs-mip.fr/),** is also represented.

Multi-model mean impacts summarized for the *ssp585* scenario in Figure 8 display a significant amplification under increased forcing with respect to the *hist* experiment (Figure S16). We compared the multi-model mean distribution of the regression coefficients values for historical and ssp585 by restricting the analysis to only the grid cells within the Amazon basin, as reported in Figure 8. To test the null hypothesis of equality between the two distributions (*hist* and *ssp585*), Mann-Whitney-Wilcoxon test of independence with a Bonferroni correction was performed. Among the climatic drivers, only *hfls* shows no significant difference between the two distributions: this is due to the fact that despite differences between the end tails of the distributions, the mean remains substantially the same. On the other hand, the null hypothesis is rejected for the other climatic drivers, with a very high degree of statistical significance (p-value less than 1e-03). In particular, a remarkable shift in both the mean and the shape of the distribution between the *hist* and *ssp585* experiment is observable for ENSO, *pr* and *rsds*, and to a less extent for *tas*.



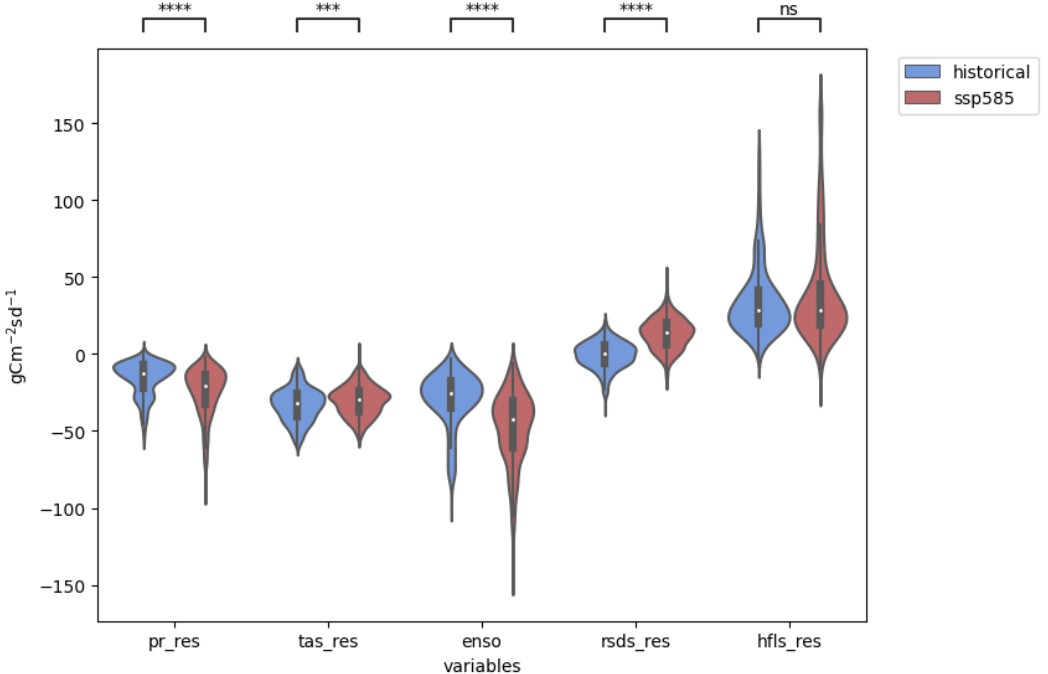

**Figure 8:** Distribution of multi-model ridge regression coefficients for predictors of Amazon NEP. Only grid-cells within the Amazon basin are included. Statistical significance, tested by means of a Mann-Whitney U-test, is reported in the stars above the plot, and refers to the following convention: *: $1.00e-02 < p <= 5.00e-02$; **: $1.00e-03 < p <= 1.00e-02$; ***: $1.00e-04 < p <= 1.00e-03$; ****: $p <= 1.00e-04$

**Discussions and Conclusions**

In this work we examined thirteen ESMs projections of Amazon basin vegetation productivity under the high radiative forcing scenario *ssp585*. Our study was motivated by the poor constraints on the Amazon carbon sink under climate change projected by CMIP6 ESM (**Brienen *et al.*, 2015; Ahlström *et al.*, 2017; Kim *et al.*, 2017; Padrón *et al.*, 2022**). First, we show that for all the ESMs except for ACCESS-ESM1-5, mean-state climatic changes by the end of the 21st century will determine a net NEP increase. Noteworthy, we attribute the projected NEP reduction in ACCESS-ESM1-5 (Figure 1a) to the implementation of Phosphorous nutrient limitation in its land module, whereas the inter-model differences observed otherwise appear to be explained by the mean-state conditions of latent heat fluxes, as well as by discrepancies in the representation of both photosynthesis and respiration processes (Figure S6). Most importantly, we expect the long-term increasing trend of shortwave incoming radiation, resulting from the reduced cloud cover over the region, and temperature, which incorporates the fertilization effect of rising atmospheric $CO_2$ concentrations (**Piao *et al.*, 2020,** Figure 5), to drive the projected increase of NEP observed in Figure 1a.



On interannual timescales, temperature, together with precipitation and ENSO, is projected to impact more strongly on NEP with respect to the historical period (Figure 8). The hotspot of these impacts is the northeastern part of the Amazon rainforest,

where stronger inter-model agreement is found, in particular for temperature and ENSO (Figure 7). The same region is also highly sensitive to latent heat fluxes, as moisture fluxes from land to the atmosphere help mitigating the adverse influence of elevated temperatures on vegetation productivity (Figure 7 and Figure S16). Thus, we can expect ENSO to play an increasing role in the variability of vegetation productivity at interannual time-scales, in accordance with previous conclusions (**Kim *et al.*, 2017**). Nevertheless, we also demonstrated that ESMs robustly show that, independently from ENSO forcing, local

processes modulated by global warming will more strongly impact vegetation productivity in the Amazon basin. These especially concerns temperature anomalies and latent heat fluxes, with respectively a negative and a positive impact on vegetation productivity.

Furthermore, we were able to identify shortwave incoming radiation and, to a lesser extent, precipitation, as the main factors to discriminate the diverse representation of NEP interannual variability across ESMs (Figure 6). This aspect could be

potentially related to the mean-state biases of precipitation and incoming shortwave radiation that all models exhibit within the Amazon basin (Figure S2 and Figure S3): a higher contribution of precipitation and a lower contribution of incoming radiation tend to inherently compensate for the respective opposite biases, promoting an increase of vegetation productivity at interannual time-scales.

We acknowledge that various factors not addressed in this research could be relevant for understanding the evolution of the

carbon sink in the Amazon basin and discriminate among ESMs behaviour, among which land-use change, disturbances (fire) and $CO_2$ fertilization are among the most prominent ones (**Padrón *et al.*, 2022**). Nevertheless, we tackled two additional hypotheses regarding the future of the Amazon carbon sink. The first hypothesis concerns an increased "lag-effect" of El Niño events, that is a prolonged impact of El Niño on vegetation productivity in the months following the peak sea-surface temperature anomalies in the equatorial Pacific. Roughly half of the models exhibit a stronger negative effect of El Niño on

Amazon NEP in the ssp585 scenario, but only ACCESS-ESM1-5 displays a prolonged effect over time (Figure S17). The second hypothesis concerns a change in the role of the Tropical North Atlantic (TNA) for Amazon vegetation productivity. After removing the ENSO influence on the TNA signal, the TNA control over the Amazon region is nonsignificant during the historical period, and very limited in the future scenario (Table SI2 and Figure S18). Our results thus indicate that lagged responses to ENSO and TNA influences do not contribute significantly to the multi-model uncertainty in projected Amazon

NEP.

Undoubtedly, the ESMs used in this study suffer from several limitations. First, the models exhibit important biases, lacking to some extent the realism and robustness in the representation of observed carbon flux variability. Most importantly, all the ESMs underestimate the Amazon NEP throughout the calendar year, with particularly evident inter-model uncertainty during the boreal winter season, as a combination of a strong overestimation of TER and underestimation of GPP (Figure S4 and

Figure S5). Second, an heterogeneous representation of vegetation and land carbon processes across ESMs terrestrial models, such as nutrient limitation, phenology, drought and heat related tree mortality, contributes to the observed spread in land carbon



sink projections **(Negrón-Juárez *et al.*, 2015; Davies-Barnard *et al.*, 2020; Koch, Brierley, *et al.*, 2021; Peano *et al.*, 2021; Padrón *et al.*, 2022)**. Additionally, drier-than-observed conditions persist in the Amazon basin ESMs climatology, possibly related to a negative bias in simulated cloud cover, leading to an overestimation of the net downward solar radiation at the

surface (Figure S2 and Figure S3). Regarding ENSO, a lower amplitude signal is observed in the ESMs: this implies a weaker eastward displacement of the Walker circulation, resulting in less anomalous precipitation amounts in the Amazon basin. Finally, our study also confirms that significant uncertainties still persist regarding how ENSO and its associated teleconnections will respond to elevated radiative forcing in the future (eg Figure S7) **(Yeh *et al.*, 2018; Fasullo, 2020; Guilyardi *et al.*, 2020; Cai *et al.*, 2021)**.

Despite these uncertainties, our research underscores the primary role of mean-state climatic changes in shaping the response of the Amazon carbon cycle under high radiative forcing conditions, regardless of changes in ENSO. Clearly, if, and by which extent, the Amazon forest will remain a carbon sink is still difficult to predict with current ESMs. Nevertheless, we reveal that current ESMs vegetation productivity evolution within the Amazon basin diverges primarily due to discrepancies in the representation of surface energy fluxes, in particular shortwave incoming radiation and latent heat fluxes. Furthermore, we

stress that incorporating Phosphorous limitation into ESMs land modules could mitigate uncertainties surrounding the future evolution of the Amazon vegetation productivity, likely leading to a reduced end-of-century carbon sink.

Therefore, not only improving the representation of ENSO dynamics and increasing the realism of surface energy fluxes within the Amazon basin will ameliorate the reliability of models projections regarding its carbon sink evolution, but most notably providing ESMs with up-to-date land processes representation, among which a mechanistic representation of plant hydraulic

stress, dynamic vegetation and phosphorous cycle, can drastically increase the comparability of the results and help discriminating the main drivers of regional and global land carbon sink anomalies.

**Code availability**: The code used to perform the analysis is publicly available at https://github.com/Matteo-Mastro/Amazon_CMIP6


**Data availability**: CMIP6 data are freely available and accessible from the ESGF repository (https://aims2.llnl.gov/search). FLUXCOM energy and carbon fluxes data are accessible from the Data Portal of the Max Planck Institute for Biogeochemistry, previous registration (https://www.bgc-jena.mpg.de/geodb/projects/Home.php). HadISST dataset is freely accessible from the MetOffice website (https://www.metoffice.gov.uk/hadobs/hadisst/data/download.html). ERA5 and ERA5-Land data are freely

accessible from the Copernicus Climate Data Store (https://cds.climate.copernicus.eu/#!/home). The Amazon shapefile used for computing the spatial mean statistics is freely available from the Amazon basin water resources observation service (https://hybam.obs-mip.fr/).

**Author contribution**: MM designed the study with contributions and feedbacks from DZ and DP. MM developed the model

code and performed the analysis. MM prepared the manuscript with contributions and feedbacks from all the co-authors.

**Competing interests**: The authors declare that they have no conflict of interest.



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
