# Peer review of "Uncertainty in Amazon vegetation productivity in CMIP6 projections driven by surface energy fluxes"

_EGUsphere, 2024_

## Author Comment (AC1)

**Reply to Reviewer 1:**

We thank the Reviewer for providing useful comments and suggestions to improve our work. We intend to address all the concerns raised by the Reviewer and refine the methodology adopted to identify the dominant factors of vegetation productivity. Specifically, the revised version of the manuscript will explicitly account for the long-term contribution effect of $CO_2$ fertilization using the C4MIP 1pctCO2-bgc simulations available for 11 out of the original 13 ESMs considered in our original study. We will also use the 1pctCO2-rad simulation to disentangle the long-term climatic effect on ecosystem productivity, thus adopting the carbon-cycle framework to understand the relative contribution of both factors to the Amazon carbon sink in a future high-radiative forcing scenario. We will therefore revisit the methodology for estimating long-term changes as described above, while maintaining the multilinear regression framework used for assessing inter-annual variability responses of ecosystem productivity. Here, we will modify the regression by following the suggestions of the Reviewer and test the effects of precipitation and alternatively soil moisture, but neglecting the contribution of latent heat fluxes. Regarding the negative correlation of precipitation with NEP, we acknowledge that this unexpected incongruence is due to an unfortunate mislabeling of the Figure 7 and Figure 8 panels, so that precipitation is associated, as expected, with a positive regression coefficient. This error comes from a wrong figure upload into the final document, and it's resolvable without additional analysis. To conclude, we are confident that our revised study will benefit from these improvements both in terms of clarity and with respect to the consistency of the methodology and reliability of the results presented.

---

## Author Comment (AC2)

**Reply to Reviewer 2:**

We thank the Reviewer for the useful comments and considerations expressed on our manuscript. We agree with the suggested changes and plan to revisit the methodology in a revised manuscript as described in the following. First, we will modify the estimation of long-term effects on ecosystem productivity in the Amazon. We won't address this research question with the use of the composite analysis anymore, specifically with the methodology adapted from Power and Delage (2018). Instead, we will adopt the carbon-cycle feedback framework by considering two additional sets of simulations from the C4MIP project, namely the 1pctCO2-bgc and 1pctCO2-rad. With the former, we aim to assess the contribution of $CO_2$-fertilization to the cumulative carbon sink, whereas the latter allows us to estimate the influence of climatic factors (at the net of $CO_2$ fertilization) on terrestrial carbon. In this way, we will disentangle the relative role of both factors for the Amazon carbon sink in a future high-radiative forcing scenario, overcoming limitations of our former analysis. Second, we will modify the estimation of vegetation productivity inter-annual variability, maintaining the original multi-linear regression approach, but neglecting latent-heat fluxes, as we acknowledge the risk of a spurious attribution if evapotranspiration correlates with GPP due to the effect of $CO_2$, which causes a closure of the stomata and thus a reduction of vegetation productivity and inhibition of plant transpiration. With this multilinear regression, we will still account for the relative contribution of ENSO when estimating vegetation productivity, but rather than focusing on the DJF season, we will consider the yearly means of the variables of interest. This will help us overcome the fact that only a subset of the ESMs correctly reproduces boreal winter as the main season affected by ENSO impacts in the Amazon basin, as already shown in Figure S17 of our manuscript (attached below). Additionally, we will compare the capability of our multi-linear model to reproduce carbon fluxes IAV by accounting for the alternative contribution of soil moisture and precipitation, as suggested by the Reviewer. Lastly, we will assess the sensitivity of our multi-linear regression model to the number of realizations used for reach model, ensuring that any potential bias from overrepresentation of certain models is controlled. With these modifications, we are confident to satisfactorily address the different factors driving terrestrial carbon fluxes within the Amazon basin, both on long-term and interannual time-scales, to ultimately provide a fully convincing revised study in terms of both methodology and interpretation.

[Figure]

**Figure S17:** Time lagged ENSO teleconnection effect on Amazon basin NEP. Displayed are the value of the regression coefficient between the standardized Nino3.4 index in the DJF season, and the 3 months running zonal NEP mean in the Amazon basin. Please note that, on the y-axes, the inverse value of NEP regression coefficient is shown for clarity (higher values depict higher negative NEP anomalies).

---

## Author Comment (AC3)

**Reply to Reviewer 3:**

We want to thank the Reviewer for providing these useful comments. In the light of this and other Reviewers' considerations, we are reassessing our approach, especially for what concerns the estimation of long-term effects on vegetation productivity. Rather than using the methodology as described by Power and Delage (2018) as in the original study, in a revised work we will estimate long-term effects with the carbon-cycle feedbacks framework, which enables us to explicitly account for the contribution of $CO_2$-fertilization by using the 1pctCO2-bgc simulations. Additionally, we will estimate the long-term climatic contribution to the Amazon carbon sink by using the 1pctCO2-rad simulation, which factors out the carbon-concentration feedback. With this respect, we will refer to the previous literature in order to provide the necessary context for our results and better clarify the novel aspects in our study. In a revised manuscript, we plan to maintain the multi-linear regression framework for describing the inter-annual variability of carbon fluxes within the Amazon basin. Despite non-linear effects are not accounted for in our multi linear regression model, we will be able to capture the biggest part of carbon-fluxes variability in the Amazon basin. We are confident this is the case for our work given that we will be considering annual-mean values, where the interaction of ENSO with local climate are more predictable and quasi-linear, as opposed to what is observed at higher frequencies (daily to seasonal timescales) and higher spatial resolutions, a condition where indeed non-linear effects could be predominant, as correctly pointed out by the Reviewer. We believe these adjustments will allow us to overcome the limitations of the original manuscript and provide a fully convincing methodology and robust results.

---

## Author Response (AR1)

**Reviewer 1:**

Mastropierro et al made the use of CMIP6 to investigate Amazon land carbon sink's uncertainties and underpinning drivers. This is a really important topic and I think they can potentially make useful contributions to the community. Nevertheless, I have several critical concerns about the methods before I can recommend for publications.

We thank Reviewer #1 for their appreciation of our work and for the useful comments. We have addressed all the Reviewer's comments and suggestions in the revised version of our manuscript. Motivated by the reviewers' comments, we have performed additional analyses and revised several aspects of the study, including improvements to the abstract and conclusions. We are confident that the revised manuscript is much stronger and more in focus than the original manuscript. Below we provide a point by point reply to the Reviewer's comments (original comments by the Reviewer in black, our response in blue).

1. Section 2.2.2. Have you removed the effects of CO2 fertilization on carbon? If not, the equations are the combined effects of eCO2 and climate impact.

We thank the Reviewer for this very important comment. In the original study, we did not explicitly remove the effects of $CO_2$ fertilization on NEP in the equations reported in section 2.2.2 of the manuscript, so that the NEP mean state changes ($\Delta MS$), reported in equation 1, were indeed the combined contribution of mean climatic changes and $CO_2$ fertilization effects. In the revised manuscript, we have disentangled these two contributions by taking advantage of C4MIP simulations: 1pctCO2-bgc, 1pctCO2-rad, ssp585-bgc and ssp585-rad. Specifically, we estimated the effect of $CO_2$ fertilization on vegetation productivity by using the biogeochemically-only coupled simulation, while we estimated the carbon-climate feedback by using the radiative-only coupled simulation.

The methodology section of the manuscript has been restructured, so that our previous section 2.2.2 "Separation of mean-state and ENSO effects" has become section 2.2 "Long-term mean-state climatic effects" in the revised version. Please refer to lines 124-143 page 5 of the revised manuscript for the revised section 2.2. These changes make the study more robust, and lead to different conclusions, specifically that $CO_2$ fertilization becomes the most prominent factor dominating uncertainties and influencing the Amazon long-term cumulative carbon sink trend across ESMs. Please refer to lines 410-414, page 18 of the revised manuscript for the updated section of the conclusions.

2. The use of water availability: soil moisture is more relevant for plant carbon uptake than precipitation. Why the authors did not choose mrso?

Thank you for bringing this point to the discussion. Given the Reviewer's suggestion and the general modifications in the methodology, in the revised version of the manuscript we consider as water availability proxy soil moisture, due to its stronger relevance for plant carbon uptake. We originally decided to adopt precipitation rather than soil moisture in equation 3 because of the direct modulation of ENSO on precipitation variability, given that the correlation between precipitation and ENSO appeared to be stronger than the one between ENSO and soil moisture. The revised version of the net biomass productivity is:

$$\Delta NBP = \frac{\delta NBP}{\delta T} * \Delta T + \frac{\delta NBP}{\delta mrso} * \Delta mrso + \frac{\delta NBP}{\delta SW_{in}} * \Delta SW_{in} + \epsilon \qquad (1)$$

Accordingly, ENSO is not anymore a factor we consider within equation 3. In the revised study, we model the effects of ENSO by considering soil-moisture and temperature-driven ENSO anomalies within the new equations 4 and 5 (shown below):

$$dNBP_{n34,T} = \frac{\delta NBP}{\delta T} * \frac{dT}{dn34} \tag{2}$$

$$dNBP_{n34,mrso} = \frac{\delta NBP}{\delta mrso} * \frac{dmrso}{dn34} \tag{3}$$

The methodology section of the manuscript has been consequently restructured, so that our previous section 2.2.3 "Effects of climatic drivers" becomes, with modifications, section 2.3 "Carbon fluxes sensitivity at inter-annual timescales". Please refer to lines 145-186 page 6 and 7 of the revised manuscript for section 2.3. These changes make the study more robust, and lead to different conclusions. In particular, we first show that GPP sensitivity to shortwave incoming radiation and heterotrophic respiration sensitivity to both soil-moisture and temperature dominate NBP divergence across ESMs at the interannual time-scale. Then, we demonstrate that temperature anomalies drive the increase in carbon sink impacts of ENSO under global warming at the inter-annual timescale. The updated conclusions are reported in line 410-440, page 18 of the revised manuscript.

3. To my surprise, why pr is negatively correlated with NEP. A drought causes more NEP, which is not feasible.

We thank the Reviewer for making us notice this incongruence. This is merely due to an unfortunate mislabeling of Figure 7 and Figure 8 panels in the original manuscript, where precipitation was negatively correlated with NEP, whereas precipitation is actually positively correlated with vegetation productivity within the area of study, as expected. The mistake comes from having uploaded into the final document a wrong version of the figure we generated. As replied to the previous comment, we decided to use soil moisture rather than precipitation as a proxy for water availability, thus the new figure does not include anymore precipitation. We report below the new Figure 5 (ex Figure 7) that shows the positive influence of soil moisture anomalies in panel b. Please refer to lines 272-343 of pages 11-14 of the revised manuscript for an overview of changes made to section 3.3 "Inter-annual variability of carbon fluxes".

[Figure]

**Figure 1:** Multi model ensemble mean of the coefficient values for the climatic drivers obtained by the multi-linear regression, for the ssp585 scenario. Hatches represent those grid cells for which at least 8 out of 11 ESMs agree in the sign of the predictor value. The Amazon basin, obtained from the SO HYBAM service (https://hybam.obs-mip.fr/), is also represented.

Small comments:

- Lines 54 -57/61-63: Water availability control on interannual variability of tropical biomes is large and could increase.

Ref:

Humphrey, V. *et al*. Sensitivity of atmospheric CO2 growth rate to observed changes in terrestrial water storage. *Nature* **560**, 628-631 (2018). https://doi.org:10.1038/s41586-018-0424-4

Liu, L. *et al*. Increasingly negative tropical water–interannual CO2 growth rate coupling. *Nature* **618**, 755-760 (2023). https://doi.org:10.1038/s41586-023-06056-x

We thank the Reviewer for suggesting these additional works that demonstrate the importance of water availability control on tropical biomes. We have rephrased the sentences accordingly and explicitly considered these works in the manuscript, as reported in lines 55-60 of the revised manuscript:

"Despite indications of temperature-driven GPP anomalies were responsible for decreased Amazonian carbon sink in the 2015/2016 event (Bastos et al., 2018; Zhang et al., 2019), it is still currently debated whether fluctuations in temperatures or water availability are the dominant drivers for interannual carbon variability of tropical biomes, with recent research indicating that the importance of water availability as a controlling factor has increased in the past decades (Jung et al., 2017; Humphrey et al., 2018; Liu et al., 2023; Zhang et al., 2023)."

- Lines 177-180. Shall we say this? The units are not same compared to NEP uncertainty. Maybe a normalized quantification of uncertainty is fair for comparisons, like spread divided by mean.

We agree with the Reviewer that a more precise quantification of uncertainty would better describe the differences among models. Therefore, following the Reviewer's suggestion, in the revised manuscript we include results from a calculation of normalized uncertainty in the projections of carbon fluxes within the Amazon basin, for the different ESMs (reported as z-score standard deviation). The additional text reads as follows (see lines 208-218 pages 7 and 8 of the revised manuscript):

"Regarding the intra-model spread, the highest influence of internal climatic variability ($\pm 1$ standard deviations) is observed in precipitation and shortwave incoming radiation, followed by soil moisture (spreads in Figure 2c,e and d). This indicates that within the Amazon basin the major source of uncertainty deriving from internal climate variability is associated with cloud formation and coverage, which is causally linked with the amount of precipitation (thus soil moisture content) and shortwave incoming radiation within the regional domain. All the carbon fluxes from which NBP is derived (GPP, Ra and Rh) depict an increasing trend throughout the 21$^{st}$ century (Figure S5), with the notable exception of GPP and heterotrophic respiration for ACCESS-ESM1-5. Among these carbon fluxes, Rh presents the highest end of 21$^{st}$ century normalized uncertainty (186.74 gCm$^{-2}$, z-score std-dev of 1.62), followed by GPP and Ra (753.59 and 548.71 gCm$^{-2}$, z-score std-dev of 1.12 and 0.76 respectively). This shows that uncertainty in cumulative NBP does not solely stem from single climatic factors; instead, it arises from inconsistencies and differences in how models represent photosynthetic activity, autotrophic and heterotrophic respiration."

Additionally, our aim here is twofold: on the one hand, we want to show that intra-model uncertainty is higher for climatic variables rather than for carbon fluxes, and on the other hand we want to show that divergence in carbon cycle projections is mostly related to terrestrial carbon sensitivity to climate, rather than climatic uncertainty itself. In the revised manuscript, we have therefore rewritten this part to emphasize

the differences among ESMs in the long-term trends of the variables considered (please see lines 191-197, page 7 of the revised manuscript):

"Considering the projections of cumulative carbon sink in Figure 1a and carbon fluxes in Figure S5, the inter-model uncertainty is much higher compared to intra-model uncertainty (which stems from the instrinsic climatic variability expressed in each realization and is represented by the ±1 standard deviation spread in Figure 1 and Figure S5). For the physical variables in Figure 1b-e, intra-model uncertainty is considerably higher than for carbon fluxes and reflects the substantial internal climate variability intrinsic in each simulation. These considerations already highlight that part of the divergence across carbon cycle predictions is related to differences in the land sensitivity to climatic forcings, rather than uncertainties in the evolution of the climate itself. Overall, the climatological variables present a stronger agreement and coherence in the sign of projected changes among ESMs with respect to NBP.".

We report the revised Figure 1 below:

[Figure]

**Figure 2:** Long-term trends of (a) cumulative NBP, (b) temperature, (c) precipitation, (d) soil-moisture, (e) shortwave incoming radiation in the Amazon basin for the historical and ssp585 experiments. Trends are computed with respect to the first 30-year mean of the historical period (1850-1880), and are visualized as a 10-year moving average for clarity. For the models with more than one realization, both the model-ensemble mean (line) and the spread (±1 standard deviation, shading) are shown.

**Reviewer 2:**

Summary:

The authors examine drivers of uncertainty in and divergence among CMIP6 model projections of Amazon carbon cycling. First, they separate the C cycle changes driven by ENSO from those driven by mean-state climate change. Then, they use regression approaches to separate the effects of mean-state changes resulting from different climate drivers, including precipitation, temperature, solar radiation, and latent heat flux. They find that changes in ENSO account for relatively little change in the Amazon C cycle relative to mean-state climate changes, which generally (but not exclusively) enhanced net ecosystem production (NEP) in the ssp585 scenario. Using the empirical approach to disentangle the various drivers of mean-state change, they further claim that most of the diversity among the models is attributable to differences in representation of C cycle responses to surface energy fluxes (solar radiation and latent heat flux in particular). Overall, while the article presents some interesting results, I think there are some major flaws that ought to be addressed.

We thank Reviewer #2 for their appreciation of our work and for the useful comments. We have addressed all the Reviewer's comments and suggestions in the revised version of our manuscript. The refined methodology led us to slightly revise the conclusions of the study, which we are confident to be now much more robust and relevant. In particular, in the revised manuscript we have explicitly considered the role of $CO_2$ fertilization, therefore being able to quantify its predominant contribution in determining long-term carbon sink trends in the Amazon basin across the considered ensemble of CMIP6 models. Additionally, we show that NBP divergence across ESMs is mainly related to the modulation of photosynthetic activity by shortwave incoming radiation and uncertainty in the representation of heterotrophic respiration sensitivity to both soil moisture and temperature. Lastly, we were able to demonstrate that temperature is the main controlling mechanism by which ENSO modulates the carbon sink at interannual timescales in the Amazon region, and that this mechanism is expected to be exacerbated by global warming, as opposed to impacts by soil-moisture anomalies.

Please see the revised abstract and conclusions in lines 8-24 and 409-440 of the revised manuscript, respectively.

General comments:

1) The biggest flaw, in my opinion, comes from the attribution of mean state change to specific drivers, especially the claim that inter-model differences in latent heat flux are driving differences in C cycling. Latent heat flux is not an exogenous variable but is instead itself driven by things like solar radiation, soil moisture, temperature and humidity as well as by plant responses to climate forcings. The authors use an empirical regression approach (Eqn. 4) to separate out the constituent drivers of NEP (including latent heat flux), but what if latent heat flux is not directly driving NEP but is instead being forced by the same drivers as NEP? In this case, both NEP and latent heat flux would be highly correlated with each other despite neither one "causing" the variation in the other but instead both just responding to a common external driver. Eqn. 4 would thus misattribute change in NEP to change in latent heat flux. I think this is very likely at least partly the case because NEP and latent heat flux are both affected by stomatal conductance (which in the models is affected by vegetation characteristics, soil moisture, VPD, etc.): all else being equal, higher conductance would increase both GPP and transpiration, and thus both NEP and latent heat flux, without latent heat flux itself causing any change in NEP. While latent heat flux could indeed cause (or at least reinforce) change in NEP through land-atmosphere feedbacks, I think the regression modeling approach used here, and thus the conclusions of the paper, are likely mistaking a correlation between latent heat flux and NEP via their common drivers for a causal driver of NEP by latent heat flux.

Thank you for this very useful comment. We acknowledge that part of the variance of NEP attributed to latent heat fluxes may be spurious due to co-variability between both variables, and we agree that this especially concerns interannual variability. To avoid spurious attribution, in the revised manuscript we use the same methodology as in the original study (i.e., ridge regression), but instead of using five independent variables in our model, we opted for using only three, namely temperature, soil-moisture and shortwave incoming radiation. We therefore neglect the role of latent heat fluxes for NBP interannual variability because of the influence of transpiration, itself influenced by stomatal conductance, on both variables. Accordingly, the revised equation 3 of the regression model is:

$$\Delta NBP = \frac{\delta NBP}{\delta T} * \Delta T + \frac{\delta NBP}{\delta mrso} * \Delta mrso + \frac{\delta NBP}{\delta SW_{in}} * \Delta SW_{in} + \epsilon \qquad (4)$$

Nevertheless, we acknowledge that latent heat fluxes may be an additional source of NBP variability at inter-annual timescales, given its impact on vapor pressure deficit, which in turn affects NBP (and we reported this in line 155 to 158 in our revised manuscript). Testing this mechanism and quantifying its effects on NBP lies outside the scope of the present research work.

Please refer to revised Section 2.3 (lines 145-169, page 5 and 6 of the revised manuscript) for details on the new methodology.

2) Related to this, elevated CO2 is almost certainly a major factor driving both change in NEP and inter-model divergence in NEP trends, but CO2 fertilization is not examined at all in this manuscript. Instead, any CO2-related changes in NEP are likely being "lumped in" with any other climate drivers that have long-term trends (like temperature). The authors claim that temperature trends positively force NEP (e.g. in fig. 5), but to me, this seems more likely to be a CO2 effect than a temperature effect. If I recall correctly (and I could definitely be wrong about this), most other studies have found that temperature on its own should *negatively*, not positively, force Amazon NEP, especially in an extreme scenario like ssp585. Because the temperature trends closely track CO2 trends in the future projections, this to me seems very likely to be a misattribution of CO2 effects to temperature effects. Previous studies (e.g. Huntzinger et al. 2017) have also shown that much of the divergence among land surface model predictions of NEP arises from differences in vegetation responses to elevated CO2.

We acknowledge the limitation of our initial analysis, as the Reviewer stresses in this comment. In the revised manuscript, we assess the Amazon ecosystem sensitivity to the increase in atmospheric carbon dioxide by analyzing the carbon cycle response to $CO_2$ according to the 1pctCO2-bgc and ssp585-bgc experiments. In these experiments, the forced increase in atmospheric $CO_2$ only affects the terrestrial and ocean ecosystems, without affecting climatic physical variables, therefore enabling us to derive the carbon-concentration feedback for the ESMs within the Amazon basin. According to this, the contribution of atmospheric $CO_2$, by means of the carbon-concentration feedback ($\beta$) on the long-term cumulative carbon sink results to be stronger than the climatic impacts ($\gamma$). Please refer to revised section 2.2 (line 124-143, page 5) of the revised manuscript for the description of the new methodology.

Because of this modification in our methodology, the results point to a predominant contribution of $CO_2$ fertilization in determining long-term carbon sink trends in the Amazon basin across the considered ensemble of CMIP6 models. Please see the detailed results on lines 235-269 of the revised manuscript.

3) A much smaller point, but for the ENSO analysis, the authors use a December-February averaging period of the Niño3.4 index but don't really justify this choice aside from saying that ENSO anomalies tend to peak in boreal winter (which is true). However, this does not necessarily imply that ENSO effects on surface climate and vegetation peak in boreal winter. Previous studies (e.g. the Zhu et al. 2017 and Zhang et al.

2019 papers already cited in the manuscript) have shown that the lags between ENSO anomalies and vegetation productivity can vary pretty substantially over the land surface and don't necessarily correspond to that DJF period. (Another more specific question I had about this was why Niño3.4 anomalies that already represented 5-month averages [line 124] were then further averaged to DJF)

Thank you for your comment on this point. In the revised version of the manuscript, we have changed the methodology to assess ENSO modulation of Amazon vegetation productivity. Specifically, we do not include anymore the Niño3.4 index in the ridge regression model, and additionally, being our focus on interannual timescales, we averaged the Niño3.4 index (as well as all other variables) to annual values. In this way, we are able to partially overcome the problem related to the lag effects of ENSO anomalies on vegetation productivity and at the same time have a clearer description of the ENSO modulation by means of temperature and soil-moisture changes. The parts of section 2.3 describing the new methodology to assess ENSO impacts are available in lines 168-186, page 6 and 7 of the revised manuscript.

This new methodological approach has led to slightly different results than the original study. Given the new equations 4 and 5 in the revised manuscript (reported below), we now identify temperature anomalies as the dominant mechanism by which ENSO impacts the vegetation carbon sink in the Amazon basin, compared to soil-moisture ENSO-driven anomalies (Figures 6 and 7, reported below).

$$dNBP_{n34,T} = \frac{\delta NBP}{\delta T} * \frac{dT}{dn34} \tag{5}$$

$$dNBP_{n34,mrso} = \frac{\delta NBP}{\delta mrso} * \frac{dmrso}{dn34} \tag{6}$$

[Figure]

**Figure 3:** Variability of NBP at interannual timescales associated to ENSO (y-axis), and mediated by either soil moisture or surface air temperature (x-axis), for every ESM. Reported are the distribution of the values within the Amazon basin, for the historical period (light-green) and the ssp585 scenario (orange). Statistical difference among the distribution of the coefficients for the two periods are tested by means of a Mann-Whitney U-test and is reported in the stars above the plots with the following convention: *: 1.00e-02 < p <= 5.00e-02; **: 1.00e-03 < p <= 1.00e-02; ***: 1.00e-04 < p <= 1.00e-03; ****: p <= 1.00e-04.

[Figure]

**Figure 4:** Only grid-cells within the Amazon basin are included. Statistical significance, tested by means of a Mann-Whitney U-test, is reported in the stars above the plot, and refers to the following convention: *: $1.00\text{e-}02 < p \le 5.00\text{e-}02$; **: $1.00\text{e-}03 < p \le 1.00\text{e-}02$; ***: $1.00\text{e-}04 < p \le 1.00\text{e-}03$; ****: $p \le 1.00\text{e-}04$

4) The authors mention that they obtained soil moisture (mrso) from the CMIP6 models, but they used precipitation in Eqn. 4. Both vegetation and soil microbial communities are likely responding more directly to soil moisture than to precipitation, so why not use soil moisture in Eqn. 4 instead of precipitation?

Thank you for the comment. We have received a similar question from Reviewer #1, and we respond here along the same line. Given the Reviewer's suggestion and the general modifications in the methodology, in the revised version of the manuscript we consider soil moisture as a proxy of water availability, due to its stronger relevance for plant carbon uptake. The reason why we originally decided to adopt precipitation rather than soil moisture in equation 3 was because of the direct modulation of ENSO on precipitation variability, given that the correlation between precipitation and ENSO appeared to be stronger than the one between ENSO and soil moisture. The revised equation 3 is:

$$\Delta NBP = \frac{\delta NBP}{\delta T} * \Delta T + \frac{\delta NBP}{\delta mrso} * \Delta mrso + \frac{\delta NBP}{\delta SW_{in}} * \Delta SW_{in} + \epsilon \tag{7}$$

Additionally, ENSO is not anymore a factor we consider within equation 3, as we model it separately by considering soil-moisture and temperature-driven ENSO anomalies as in the new equations 4 and 5 (shown below):

$$dNBP_{n34,T} = \frac{\delta NBP}{\delta T} * \frac{dT}{dn34} \tag{8}$$

$$dNBP_{n34,mrso} = \frac{\delta NBP}{\delta mrso} * \frac{dmrso}{dn34} \tag{9}$$

The methodology section of the manuscript has been restructured, so that our previous section 2.2.3 "Effects of climatic drivers" becomes, with modifications, section 2.3 "Carbon fluxes sensitivity at inter-annual timescales". Please refer to lines 145-191 page 6 of the revised manuscript for the revised section 2.3. These changes make the study more robust, and lead to different conclusions than the original study. In particular,

we first show that GPP sensitivity to shortwave incoming radiation and heterotrophic respiration sensitivity to both soil-moisture and temperature dominate NBP divergence across ESMs at the interannual time-scale. Second, we demonstrate that ENSO-driven temperature anomalies drive the increase in carbon sink impacts of ENSO under global warming at the inter-annual timescale. Conclusions are reported in line 409-440, page 18 of the revised manuscript".

5) In general, I was a little confused about section 3.4 and the corresponding methods section (2.2.3). There seem to be some inconsistencies in how the regression coefficients were described in the text and how they appear in the figures. For example, line 332 says that the models show a negative influence of temperature, but doesn't Fig. 5 show a *positive* influence of temperature? I also found the precipitation response confusing… the authors variously describe precipitation as being negatively correlated with NEP (lines 341-342) on the one hand, but also say that places with larger negative trends in precipitation have lower NEPs (lines 319-320). In order for a negative trend in precipitation to result in lower NEP, that would mean that NEP and precipitation would have to be *positively* (not negatively) correlated with each other, right? I'm also struggling to figure out how precipitation would be negatively correlated with NEP (or have a negative regression coefficient with NEP, as in fig. 5 and 6) in the Amazon.

Thank you for making us notice this unfortunate incongruence and for stressing out a lack of clarity in section 3.4. Figure 5 of the original manuscript has been removed in the revised manuscript, due to the changes in the methodology reported in our response to comment nr. 4. Additionally, in the original manuscript we mistakenly reported the coefficients of precipitation as being negative in Figure 7 and Figure 8. This was due to an unfortunate mislabeling of Figure 7 and Figure 8 panels: precipitation is actually positively correlated with NEP within the area of study, as expected. The error came from having uploaded into the final document a wrong version of the figure. New Figure 4 and Figure 5 in the revised manuscript (reported below) provide correct values and signs of the regression coefficients.

[Figure]

**Figure 5:** Partial derivatives explaining the contribution of temperature (a), soil moisture (b) and shortwave incoming radiation (c) to interannual NBP, averaged across the Amazon basin. The black vertical bars represent the spread in the predictors coefficients for models with more than one realization available, whereas the stars indicate the level of significance (p-value), averaged over the Amazon basin, associated to every coefficient. Statistical significance refers to the following convention: *: 1.00e-02 < p <= 5.00e-02; **: 1.00e-03 < p <= 1.00e-02; ***: 1.00e-04 < p <= 1.00e-03; ****: p <= 1.00e-04

[Figure]

**Figure 6:** Multi model ensemble mean of the coefficient values for the climatic drivers obtained by the multi-linear regression, for the ssp585 scenario. Hatches represent those grid cells for which at least 8 out of 11 ESMs agree in the sign of the predictor value. The Amazon basin, obtained from the SO HYBAM service (https://hybam.obs-mip.fr/), is also represented.

Specific comments:

Line 69: is "given" supposed to say "driven"?

Correct, we meant that precipitation decline is driven by increased $CO_2$ as a consequence of stomatal closure. We have corrected the sentence in the revised manuscript accordingly.

Line 75: Another ENSO property that could influence the Amazon carbon cycle is the spatial location of SST anomalies in the tropical Pacific (i.e., central Pacific vs. eastern Pacific ENSO events). Recent frequencies of central Pacific ENSO events are considerably above the norm based on paleoclimate reconstructions (Freund et al., 2019) and have been shown to have distinct impacts on GPP and NEP in many regions, especially in the Amazon (Dannenberg et al., 2021). Further, some studies have shown that central Pacific El Niños are projected to be more frequent under 21[st] century warming (e.g., Shin et al., 2022), though some of this might result from biases in CMIP6 model simulations of equatorial zonal flows (Wang & Lin, 2023). Still, might be worth noting that this (along with increases in ENSO amplitude and vegetation sensitivity to ENSO) is another potential source of future change in the Amazon-ENSO connection that is not examined here since the Niño3.4 region mixes the two (but is mostly centered in the central Pacific). To be clear, I'm not necessarily suggesting that the authors redo analyses to explicitly examine the CP vs. EP ENSO effect on Amazon C cycling, though it could be really interesting at least for future work!

Yes, we agree with the Reviewer that previous research has demonstrated the different influence of CP vs EP ENSO impacts on vegetation carbon fluxes. However, we decided to maintain the current approach based on Nino3.4 because it is the ENSO index which shows the highest correlation with climatic anomalies in various regions of the globe. Furthermore, given the bias that ESMs display in the variability of SST in the equatorial Pacific (as mentioned by the Reviewer and shown in Figure S1 of the Supplementary Material), considering the diversity of CP and EP ENSO events would result in additional uncertainty and complexity in our analysis. However, we appreciate these considerations from the Reviewer, and we decided to include them in the Discussion section of the revised manuscript (lines 361-367 page 14).

Line 92: Could using a variable number of realizations from each model cause some model overrepresentation that would bias results? For example, if Model A has five realizations but Model B only has one, then Model A would be weighted five times more than Model B in the presentation of inter-model

results. For this reason, some previous studies have only used one realization per model (Diffenbaugh et al. 2018).

It is true that using a variable number of realizations for each model results in an overall overweighting of those models with more than one realization, when the ensemble analysis is performed on multi-model averaging of all available realizations. However, this is not the approach in our study, as we always keep models separated. In our methodology, where multiple realizations are available (i.e. in the estimation of the drivers of interannual variability), the results are always averaged by model when reported in figures and therefore represent a way to highlight the forced component of variability, reducing the uncertainties related to internal climatic variability within each ESM. This is seen, as an example, in Figure 4 and Figure S12 of the manuscript for the average values and the uncertainty range (for those models with more than one realization available) of the partial derivatives of the ridge regression model.

Lines 128-129: 10th and 90th percentiles seem like pretty strict definitions. That would result in only 6 events per 60 year period. Would 20th and 80th percentiles be a better choice? I don't feel strongly about this though, so I'll leave that to the authors' discretion.

We agree with the Reviewer that the choice of the 10th and 90th percentiles may lead to few events being selected. In the revised version of the manuscript, we have changed our methodology with respect to ENSO, so that we do not rely anymore on the composite analysis (and consequently on the choice of 10th and 90th percentiles) presented in the original manuscript. Indeed, we now consider ENSO impact on vegetation productivity by means of a regression analysis (see equations 4-5 of the revised manuscript), hence we consider the whole distribution of the processes of interest. Please see the results of this new approach on line 354-406 of the revised manuscript.

Line 148: I'm not sure I understand what the authors mean by "uniquely standardized"... could you explain a little bit more how MLR-trend and MLR-iav were obtained?

Thank you for this comment. In the revised version of the manuscript, for the estimation of the long-term carbon fluxes trends we do not use anymore the regression framework specified as "MLR_trend", but use instead the carbon-cycle feedbacks framework to derive the long-term effects of $CO_2$ fertilization and climate (section 2.2 of the revised manuscript). We keep the previous "MLR_iav" methodology (which now we describe as "Carbon fluxes sensitivity at inter-annual timescales", in section 2.3), which aims to quantify the contribution of different drivers of carbon fluxes interannual variability. Therefore, we now adopt only one multivariate regression framework, in which we detrend the dependent variable, and both detrend and standardize the three independent variables. This is made clear in the revised manuscript, please check lines 151-165 page 6, where we write: "The dependent variable (NBP, as well as GPP, Ra and Rh) has been linearly detrended to remove the influence of $CO_2$ fertilization and other long-term climatic changes, while the three independent variables have been detrended and standardized (divided by their standard deviation) to obtain comparable coefficients. Therefore, the standardized coefficients represent the quantitative contribution of temperature, soil-moisture and shortwave incoming radiation considering the simultaneous confounding impacts of the other variables."

Lines 160-164: I'm confused about equations 5-8. What are the alphas in these equations? Are those alphas different for each of equations 5-8 (in other words, removing the effect of Nino3.4 on each variable individually)? If so, the nomenclature is kind of confusing.

We agree that the nomenclature was potentially confusing, because we used the same symbology ($\alpha$) to represent the different residuals of the independent variables with respect to ENSO. Nonetheless, because of the revised methodology, we disregard equations 5-8, hence the alphas reported there are not considered anymore in our current methodology.

Lines 165-169: Did the authors examine the predictive skill of their model? It could have significant regression coefficients without being a particularly skillful model. I think it would be good to do so using data withheld from calibration of equation 4.

We thank the Reviewer for bringing to the discussion the predictive skill of our model. We indeed have tested the skills of our model, for both the historical period and the ssp585 scenario, by adopting a 5-fold cross-validation process for our ridge regression model, a procedure we used with the aim to optimize the learning performance of our model itself.

The results of the predictive skills of our regression model with the 5-fold cross-validation are reported below in new Figure S10 of the Supplementary Information. In the revised manuscript, the predictive skills and the performance of the ridge regression approach are described as follows (lines 278-290, page 11):

"The skills of the model optimized with the 5-fold cross-validation procedure are reported in Figure S10. Overall, a multi-model mean coefficient of determination of 0.55 is obtained, despite with substantial differences across ESMs. Remarkably lower values of variance explained ($R^2<0.4$) are found for CESM2, CMCC-ESM2 and NorESM2-LM, whereas CanESM5, MIROC-ES2L and UKESM1-0-LL stand out as the models with the highest goodness of fit and predictive capability ($R^2>0.6$), (Figure S10a). Undoubtedly, we acknowledge the limitations that arise from the adoption of a linear ridge regression model. As we didn't account for interactive and non-linear effects among the predictors influencing NBP IAV, we are not able to capture a portion of the unexplained variance in our regression model. The results in Figure S10 reflects this fact, suggesting that other factors, as well as the effects emerging from the interaction of temperature, soil-moisture and shortwave incoming radiation, are likely having an important influence on NBP IAV, especially for CESM2, NorESM2-LM and partly CMCC-ESM2. Nevertheless, we opted for this modelling framework, as it still allows our results to be compared with prior works (Jung *et al.*, 2017; Humphrey *et al.*, 2018)."

[Figure]

**Figure S7:** Predictive skills of the ridge regression model by adopting the 5-fold cross-validation procedure, for every ESM and for both the historical and the ssp585 scenario. In panel a) is reported $R^2$, in panel b) the Explained Variance, in panel c) the RMSE and in panel d) the MAE.

Line 228: It's not clear to me what "stronger inhibition of the tropical teleconnection pathway" means.

In this sentence, we refer to the mechanism(s) by which ENSO controls temperature and precipitation over South America. More specifically, we refer to the possible increase in the frequency and magnitude of the positive phase of ENSO (El Niño) influence, with consequent stronger inhibitions in the southward shifts of the Inter Tropical Convergence Zone (ITCZ) to the Amazon basin. In the revised manuscript (see lines 329-332), we have rephrased the quoted text as follows:

"For example, several studies indicate the possibility of an increase in the frequency or magnitude of El Niño and La Niña events under global warming (Cai *et al.*, 2014, 2015; Berner *et al.*, 2020; Brown *et al.*, 2020; Fredriksen *et al.*, 2020). This strengthening of ENSO variability could have important implications for the Amazon ecosystem especially due to a stronger inhibition, under El Niño events, of the boreal winter southward shift of Inter Tropical Convergence Zone (ITCZ) that regulates the ENSO tropical teleconnection pathway.".

Lines 228-232: Just to clarify, is this based on a comparison of the historical ESM SST simulations to those actually observed with measurements?

Correct, we compare the ESMs SST historical simulations of ENSO amplitude to the SST data from the HadISST dataset.

Lines 274-278: This last paragraph of section 3.3 seems to give pretty short shrift to the results. Can you expand?

We have substantially modified the results section, and the paragraph mentioned by the Reviewer is now in section 3.2 "Long-term carbon sink sensitivity", which can be found at lines 228-262, pages 9 and 10 of the revised manuscript.

Lines 282-284: I'd suggest being more explicit here. What does "satisfactory" mean? What skill thresholds need to be met for it to be satisfactory? Was the assessment of model skill based on data withheld from model training?

We agree with the Reviewer that we need to remain confined within an objective quantification for our regression model skills. In the revised manuscript we avoid the use of vague terms such as "satisfactory" and only describe the performance of the model in terms of $R^2$, RMSE, MAE and EVS.

The text added in the revised manuscript (see lines 278-290) can be found above, as the reply to the Reviewer's specific comment about lines 165-169.

Fig. 1 (also worth checking the other figures as well): some of the text (especially in the legend) is very hard to read.

We thank the Reviewer for making us notice this: we have updated the legend and the labels of the figures in the revised manuscript and in the supplementary information, carefully checking for readability and completeness of information.

Fig. 2: Is it possible to include confidence intervals on these?

Yes, it is possible to include confidence intervals for those ESMs presenting more than one realization. To do so, Figure S18 of the revised manuscript (ex Figure 2 of the original manuscript) now illustrates confidence intervals of the Nino3.4 standard-deviation, which represent the ENSO amplitude values for those models with more than one realization available. Updated Figure S18 is reported below:

[Figure]

**Figure S88:** ENSO amplitude change as represented by the Nino3.4 signal standard deviation from the historical period (green crosses) to the future ssp585 scenario (orange dots). The black cross represents the value of Nino3.4 signal amplitude calculated from the HadISST dataset.

**Reviewer 3:**

The authors explored the inter-model diversity of vegetation productivity simulated under CMIP6 historical and SSP585 scenarios, and found out that the uncertainty in Amazon vegetation productivity in CMIP6 projections is driven by the dominant role of local mean-state climate changes and the minor role of El Nino-Southern Oscillation (ENSO). In particular, the surface energy balance components (shortwave incoming radiation & latent heat fluxes) are the main cause of divergence actress ESM responses. They also pointed out the need for phosphorus limitation.

We thank the Reviewer for the appreciation of our work and for the useful comments that address key points of the manuscript. Motivated by the reviewers' comments, we have performed additional analyses and revised several aspects of the study, up to improving abstract and conclusions. We are confident that the revised manuscript is much stronger and more in focus than the original manuscript. Below we provide a point by point reply to the Reviewer's comments (original comments by the Reviewer in black, our response in blue).

Major comments:

1. Novelty

This paper investigates the drivers of inter-model diversity, with a focus on local mean climate and ENSO, introducing novelty in its findings. In particular, this paper goes beyond coarse climate but examines specific climate drivers and their effects on inter-model variability.

However, some statements appear misleading, implying that it's a new discovery, when it might not be. For example, in the result section (3.1), the authors state that "Inter-model uncertainty is much higher than intra-model uncertainty, originated by ESMs internal climate variability" (Line 176) and "This shows that uncertainty in NEP does not solely stem from photosynthesis or respiration; instead, it arises from inconsistencies and limitations in how models represent both processes" (Line 197). In fact, Heavens et al (2013) already explained that ESM predictions differ because 1) models do not agree on the details of how climate will change; and 2) land carbon models are differently sensitive to the four processes: a. varied vegetation growth in fixing carbon; b. climate change driving changes in precipitation, which drives changes in vegetation growth; c. warming climate increasing microbial respiration; and d. carbon fixation slowing as vegetation deplete soil nutrients. I think it is ok to include those in the result section. However, the authors need to make it clear that these findings are not novel and align with previous studies to provide the necessary context.

We want to thank the Reviewer for this major comment. We agree that some of the results presented in our manuscript expand and confirm previous studies. Their inclusion seems to be relevant because the results stem from different models and different approaches and assumptions. Indeed, several works have examined the sensitivity of tropical terrestrial carbon cycle (including the Amazon basin) to different climatic drivers, as well as the divergence of different ESMs. However, ours is the first study, to our knowledge, to assess carbon cycle dynamics in the Amazon basin using a multi-model ensemble of CMIP6 ESMs, considering the high radiative forcing ssp585 scenario and considering within the same framework both long-term tendential changes and interannual variability. The unique use of data, approach and focus of our study requires that we provide the necessary context, hence including some results that have little novelty. In the revised version of the manuscript, we pay special attention to distinguishing results that provide new knowledge and results that confirm previous knowledge. In particular, the revised manuscript includes the following statements that put our results better in the frame of previous knowledge:

Lines 55-60: "Despite indications of temperature-driven GPP anomalies were responsible for decreased Amazonian carbon sink in the 2015/2016 event (Bastos *et al.*, 2018; Zhang *et al.*, 2019), it is still currently debated whether fluctuations in temperatures or water availability are the dominant drivers for interannual carbon variability of tropical biomes, with recent research indicating the increased importance of water availability as a controlling factor in the past decades and with ESMs failing to reproduce this observed behavior (Jung *et al.*, 2017; Humphrey *et al.*, 2018; Liu *et al.*, 2023; Zhang *et al.*, 2023)."

Lines 229-232: "Overall, inconsistencies in projected Amazon NBP cannot be simply understood as a consequence of discrepancies in the climatic factors affecting vegetation, as already discussed by Heavens et al., (2013)"

Lines 311-316: "In Figure 5, seven out of eleven models project temperature as the first NBP predictor, while for CanESM5, IPSL-CM6A-LR and MPI-ESM1-2-LR soil moisture results to be more important, and MIROC-ES2L predicts shortwave incoming radiation as the most dominant driver of NBP IAV. Overall, these results complement and partially confirm what shown in a recent research (Padrón *et al.*, 2022), which found temperature to be more important than soil moisture to explain interannual variability of NBP in an ensemble of models, considering the ssp126 low emission scenario."

Lines 375-378: "While the carbon cycle response to ENSO within the Amazon region presented here is well in agreement with previous research (Kim *et al.*, 2017; Betts *et al.*, 2020; Le *et al.*, 2021; Le, 2023), our results further identify temperature as the key factor in the mechanism by which ENSO affects Amazon carbon fluxes in a high radiative forcing future scenario."

Lines 420-433: "Specifically, we identified $CO_2$ fertilization as the predominant mechanism determining the long-term Amazon carbon sink trend and uncertainty under the range of atmospheric $CO_2$ concentrations of the ssp585 scenario (400-1135 ppmvv). These results reaffirm, for the CMIP6 generation of ESMs, what reported in a previous study by Huntzinger et al. (2017), that under different research assumptions showed the predominant role of vegetation sensitivity to $CO_2$ in shaping the net carbon sink variability across CMIP5 generation Land Surface Models (LSMs). Additionally, we disentangled the fundamental physical processes behind net carbon sink discrepancies across ESMs, highlighting dominant mechanisms affecting simulated carbon fluxes uncertainty for the Amazon basin ecosystem. Particularly, we show that the ensemble divergence of NBP under future warming scenario is largely determined by GPP modulation by shortwave incoming radiation and uncertainty in the representation of heterotrophic respiration sensitivity to both soil moisture and temperature. Our multi-model ensemble approach expands the results obtained in previous researches (Ma *et al.*, 2021) by allowing for an explicit consideration of model uncertainty. By showing the strong uncertainties in driving factors of heterotrophic respiration, we suggest that not only ESMs differ in the positive modulation of temperature on Rh (controlled by the $Q_{10}$ equation), but also largely disagree in the association between soil moisture and soil decomposition rates leading to respiration fluxes, as recently pointed out by Guenet et al., (2024)."

2. Data, Method and Results

I have major concerns in this part. The method section lacks clarity/validity and the results are not consistent.

We thank the Reviewer for highlighting issues with the description of the methodology in our original study. As a preliminary general answer to the Reviewer's concerns, in the revised manuscript we have restructured the methodology to better address long-term mean state climatic effects, including $CO_2$ fertilization, and

the influence of different climatic factors on carbon fluxes at interannual timescales. Consequently, below are the replies to the three specific concerns raised by the Reviewer.

First, the way to define El Nino and La Nina might not be convincing. The authors defined El Niño and La Niña events using the 90th and 10th percentiles of the DJF averaged Nino3.4 index time series. Citations or evidence were missing to prove why they chose 90/10th percentiles as thresholds. They claimed that they detrended the average Nino 3.4 index over Dec-Feb, by means of a 1st order polynomial and normalized (without citation and proof as well). At least they need to clarify how "the means of a 1st order polynomial" were defined and which normalization was applied (eg., Z-score normalization, or Min-max scaling). If Z-score normalization was used, I would recommend using standard deviations as thresholds to define El Niño and La Niña events, instead of percentiles.

We agree with the Reviewer that the choice of the $10^{th}$ and $90^{th}$ percentiles is subjective. In the revised version of the manuscript, we have changed our methodology with respect to ENSO, so that we do not rely anymore on the composite analysis, hence on the $10^{th}$ and $90^{th}$ percentiles definitions. Furthermore, our predictor for ENSO is a detrended time series of the canonical Nino3.4 index, aggregated from monthly to annual time resolution. We consider this an appropriate choice considering the focus on ENSO modulation on interannual timescales. The new methodology regarding ENSO is described in the revised manuscript in lines 172-191, pages 6 and 7, which read as follows:

"Given the predominant modulation of carbon fluxes IAV within the Amazon basin by means of ENSO (Mcphaden *et al.*, 2021a), a further analysis is conducted to assess the ENSO contribution to Amazon vegetation productivity mediated by either temperature and water-availability (using soil moisture as a proxy). For this, an annual time series of ENSO is obtained for each historical and ssp585 realization by averaging the corresponding monthly Nino3.4 index over the calendar year. The Nino3.4 index is defined as the 5-month moving average of mean sea-surface temperatures over the region 170-120°W and 5°S-5°N, subsequently detrended by means of a 1st order polynomial to remove the warming trend of SST. By taking the univariate sensitivities of temperature and soil-moisture to ENSO, and accounting for the partial derivatives of NBP with respect to temperature and soil-moisture as in Equation 3, we estimate the contribution of ENSO driven by the two mechanisms as below:

$$dNBP_{n34,T} = \frac{\delta NBP}{\delta T} * \frac{dT}{dn34} \tag{10}$$

$$dNBP_{n34,mrso} = \frac{\delta NBP}{\delta mrso} * \frac{dmrso}{dn34} \tag{11}$$

Using the partial derivatives $\frac{\delta \text{NBP}}{\delta \text{T}}$ and $\frac{\delta \text{NBP}}{\delta \text{mrso}}$ ensures that the effects of temperature and soil moisture are considered independently, accounting for potential confounding influences from each other, a condition not met if the univariate estimates $\frac{dNBP}{dT}$ and $\frac{dNBP}{dmrso}$ were applied. A Mann-Whitney U-test of independence with Bonferroni correction was used to assess whether the zonal values of the regression coefficients within the Amazon basin are significantly different between the historical period and ssp585 scenario. To mitigate the risk of overstating the significance of the statistical tests conducted, we employ a false discovery rate (FDR) control method based on (Wilks, 2016). This approach effectively addresses the issue of multiple hypothesis testing, ensuring a more accurate interpretation of the obtained results."

Because of our new methodology, the revised manuscript results and conclusions change in two aspects. First, we do not dtermine anymore mean-state changes by using ENSO composites, thus using the criticized $10^{th}$ and $90^{th}$ percentiles. By the new analyses, we identify $CO_2$ fertilization as the main factor influencing long-term cumulative carbon sink evolution in the Amazon basin (see the next reply for a detailed

description of this). Second, equations 4 and 5 reported above allow us to determine that temperature is the main controlling variable involved in the mechanism by which ENSO modulates the carbon sink in the Amazon region at interannual timescales, and that this mechanism is expected to be strongly exacerbated by global warming, as opposed to soil-moisture anomalies impacts. We report this conclusion in lines 433-438 page 18 of the revised manuscript: "Additionally, our results point towards a stronger ENSO-driven temperature-mediated impact on carbon sink anomalies for the vast majority of ESMs, compared to ENSO-driven impacts associated with deficits in water availability. Accordingly, the considered CMIP6 multi-model ensemble shows a robust and statistically significant increase in carbon sink sensitivity to ENSO-driven temperature anomalies under global warming in the region of the Amazon basin: as a consequence, climate change is likely to significantly diminish the Amazon ecosystem capacity to function as a carbon sink, further aggravating the atmospheric $CO_2$ burden."

Second, the separation of mean-state climate and ENSO effects might be confusing. According to Power and Delage 2018 (the method the authors applied), El Nino effects on climates are defined as $\boldsymbol{\Delta EN}$=($\boldsymbol{\delta ENssp}$)-($\boldsymbol{\delta ENhist}$)=(Essp − Nssp)-(Ehist-Nhist), where ESSP denotes climates averaged over the El Nino events under SSP scenarios and Nssp denotes the climates averaged over the neutral years under SSP scenarios. Notably, the definition in Power and Delage (2018) is about the effects on climate, rather than effects on NEP. The author should make it clear that the effects of ENSO on climate differ from its effects on NEP. They need explain how they applied $\boldsymbol{\Delta EN}$ to get the impacts of El Nino on NEP(Fig. 3&4). If they directly used the idea of $\boldsymbol{\Delta EN}$ to calculate effects on NEP— where ESSP and Nssp denoted NEP averaged over the El Nino events and over neutral years under SSP scenarios, respectively, $\boldsymbol{\Delta MS}$ (=$Nssp − Nhist$) would be the long-term trend effects, including CO2 fertilization. The inclusion of CO2 fertilization effects can largely lead to misinterpretation of the results and may explain conflicted findings presented in the paper.

For example, Fig. 4A demonstrates that both climate and La Niña have positive impacts on NEP, which seems counterintuitive. Rising temperatures and water deficits are expected to increase stress on vegetation, and La Niña, often associated with flooding, should exhibit some negative effects.

We acknowledge that the framework used by Power and Delage to derive mean state changes is originally applied to climate conditions, rather than vegetation productivity. By applying it to vegetation productivity within the Amazon basin, we incorporated the $CO_2$-fertilization effect, which we agree was a substantial limitation of our original approach. In the revised manuscript, we overcome this limitation by disentangling the $CO_2$-fertilization effect and the climate impact in the estimation of long-term effects on the cumulative Amazon carbon sink. We did this by taking advantage of C4MIP simulations, specifically the 1pctCO$_2$-bgc, 1pctCO$_2$-rad, ssp585-bgc and ssp585-rad simulations with the goal of estimating the effect of $CO_2$ fertilization on vegetation productivity (by using the biogeochemically only coupled simulation), as well as the carbon-climate feedback (by using the radiative-only coupled simulation) in the Amazon basin. With this methodology, we are therefore able to distinguish these two different mechanisms affecting ecosystem carbon fluxes in the region. This new methodology can be found in lines 124-143, page 5 of the revised manuscript:

"Increasing trends of atmospheric $CO_2$ concentrations affect terrestrial carbon sinks directly through a fertilization effect on vegetation (carbon-concentration feedback) and indirectly by forcing changes in the physical climate via a strengthened greenhouse effect, which in turn affects vegetation (carbon-climate feedback). A common approach to disentangle the two effect relies on the carbon-cycle feedback framework, by which it is possible to estimate the magnitude of the carbon-concentration feedback and the

carbon-climate feedback (Jones *et al.*, 2016; K. Arora *et al.*, 2020). The contribution of these two feedbacks is estimated with the following equations:

$$\beta = \frac{\Delta NBP_{cum,BGC}}{\Delta ppm_{BGC}} \tag{12}$$

$$\gamma_T = \frac{\Delta NBP_{cum,RAD}}{\Delta T_{RAD}}; \quad \gamma_{mrso} = \frac{\Delta NBP_{cum,RAD}}{\Delta mrso_{RAD}}; \quad \gamma_{SW_{in}} = \frac{\Delta NBP_{cum,RAD}}{\Delta SWin_{RAD}} \tag{13}$$

Carbon sink is represented here by cumulative NBP, whose long-term sensitivity to $CO_2$ ppmv ($\beta$), and climate ($\gamma$), is estimated from biogeochemically only coupled simulations (1pctCO$_2$-bgc and ssp585-bgc) and radiative only coupled simulations (1pctCO$_2$-rad and ssp585-rad) respectively. Within the former, the increased $CO_2$ atmospheric concentration is uniquely exerting a biogeochemical effect which interests the terrestrial and ocean carbon cycle processes, whereas in the radiative simulation only the radiative transfer processes in the atmosphere are affected by the changing atmospheric $CO_2$ concentrations, with no consequences for biochemical processes (Friedlingstein *et al.*, 2006; Jones *et al.*, 2016). Changes are expressed with respect to the first year of the simulations; for 1pctCO2 experiments only the years with atmospheric CO2 ranges similar to the ssp585 scenario (400-1135 ppm, resulting in 104 years) have been considered. On the other hand, the carbon-climate feedback has been further computed with respect to surface atmospheric temperatures (T), soil moisture (mrso) and shortwave incoming radiation (SWin), to represent the variety of mean-state climatic changes affecting the cumulative carbon fluxes within the Amazon basin."

Consequently, without the need to adopt the framework of Power and Delage (2018), we were able to identify $CO_2$ fertilization as the main factor influencing long-term cumulative carbon sink evolution in the Amazon basin. Specifically, the considered ensemble of ESMs projects an strengthening of the carbon sink resulting from the effect of atmospheric $CO_2$ concentrations from the ssp585 scenario, without which the carbon climate-feedback would lead to a sustained reduction in the carbon sink capacity. We report this conclusion in lines 412-416, page 18: "Specifically, we identified $CO_2$ fertilization as the predominant mechanism determining the long-term Amazon carbon sink trend and uncertainty under the range of atmospheric CO2 concentrations of the ssp585 scenario (400-1135 ppmv). These results reaffirm, for the CMIP6 generation of ESMs, what reported in a previous study by Huntzinger et al. (2017), that under different research assumptions showed the predominant role of vegetation sensitivity to CO2 in shaping the net carbon sink variability across CMIP5 generation LSMs."

Third, the method section 2.2.3 (Effects of climatic drivers) is very confusing. In this section, the authors used detrended climate anomalies to disentangle their effects on NEP from ENSO. But in the result section 3.4.1, they claimed that it is the long-term changes effects on NEP by using method 2.2.3. I have two questions here: 1) what are the different purposes between methods of 2.2.2 and 2.2.3? It appears to me that the two method sections are unrelated to each other, especially in the result presentations. If detrending climates in method section 2.2.3 is to remove the long-term trend effects (eg., partially remove CO2 fertilization), why did the authors use the method section 2.2.2, including CO2 fertilization? 2) Why did the authors apply a simple linear regression model in 2.2.3? To my understanding, El Niño events tend to interact with climate in impacting NEP, suggesting additive non-linear negative effects once climate tipping thresholds are exceeded. Although the linear model in the paper used the residuals of the single-climate regression model against ENSO, there is still lack of consideration of their interactive, non-linear effects. In this case, the method section 2.23 may not have fully eliminated the CO2 fertilization effects and may

have overlooked the interactive effects between ENSO and climate, which can cause conflicted results as well.

We agree that the methodology in our original study could generate confusion. We are confident that the revised manuscript presents a clearer and more straightforward approach. The previous sections 2.2.2 and 2.2.3 have now become section 2.2 "Long-term mean-state climatic effects" and 2.3 "Carbon fluxes sensitivity at inter-annual timescales" respectively. We therefore now clearly differentiate long-term drivers ($CO_2$ fertilization effect and mean-state climatic changes) from climatic influences at the inter-annual timescales, and this methodological framework is now reflected in the results sections 3.2 (long-term effects) and 3.3 and 3.4 for the results regarding the drivers of interannual variability of carbon fluxes. About the Reviewer's second question, we agree and acknowledge the limitation of adopting a linear regression framework in our methodology, as interactions and non-linear terms are not accounted for. Nonetheless, we opted for a state-of-the-art regression model, as the 5-fold cross-validation ridge regression that would still allow our results to be compared with prior works (Jung et al., 2017 and Humphrey et al., 2018, have used this same approach). We acknowledge this limitation in the revised manuscript (Lines 282-288):

"Undoubtedly, we acknowledge the limitations that arise from the adoption of a linear ridge regression model. As we didn't account for interactive and non-linear effects among the predictors influencing NBP IAV, we are not able to capture a portion of the unexplained variance in our regression model. The results in Figure S10 reflects this fact, suggesting that other factors, as well as the effects emerging from the interaction of temperature, soil-moisture and shortwave incoming radiation, are likely having an important influence on NBP IAV, especially for CESM2, NorESM2-LM and partly CMCC-ESM2. Nevertheless, we opted for this modelling framework, as it still allows our results to be compared with prior works (Jung *et al.*, 2017; Humphrey *et al.*, 2018)."

For instance, Fig. 5 shows that precipitation has negative effects while temperature has positive effects, which seems counterintuitive. Surprisingly, Fig. 7 and 8 both show temperature has negative effects, which conflicted with the results in Fig. 5. It seems that all these figures come from the method section 2.2.3.

I hope the authors can take these conflicted results carefully and report them with consistency if revision is invited.

Regarding the incongruences of Figures 5, 7 and 8, please note that Figure 5 in the original manuscript has been removed from the revised manuscript, following the changes in the methodology described above. We nonetheless remark here that in the original Figures 7 and 8 the negative coefficients shown for precipitation were regrettably due to an unfortunate mislabeling of the panels. In fact, precipitation is associated with a positive regression coefficient within the area of study, as expected. This is illustrated in the Figure 4 and Figure 5 of the revised manuscript (which correspond to the previous Figures 7 and 8), reported below. All figures and captions have now been carefully checked in the revised manuscript.

[Figure]

**Figure 9:** Partial derivatives explaining the contribution of temperature (a), soil moisture (b) and shortwave incoming radiation (c) to interannual NBP, averaged across the Amazon basin. The black vertical bars represent the spread in the predictors coefficients for models with more than one realization available, whereas the stars indicate the level of significance (p-value), averaged over the Amazon basin, associated to every coefficient. Statistical significance refers to the following convention: *: $1.00e\text{-}02 < p <= 5.00e\text{-}02$; **: $1.00e\text{-}03 < p <= 1.00e\text{-}02$; ***: $1.00e\text{-}04 < p <= 1.00e\text{-}03$; ****: $p <= 1.00e\text{-}04$

[Figure]

**Figure 10:** Multi model ensemble mean of the coefficient values for the climatic drivers obtained by the multi-linear regression, for the ssp585 scenario. Hatches represent those grid cells for which at least 8 out of 11 ESMs agree in the sign of the predictor value. The Amazon basin, obtained from the SO HYBAM service (https://hybam.obs-mip.fr/), is also represented.

3. Clarity

The writing in the method section can be improved and please ensure consistency between the methods and results.

We have carefully revised the writing of the methods section and made it consistent with the results presented in the revised manuscript. Please refer to the revised manuscript in lines 124-186, pages 5 to 7, for the revised methodology.

Specific comments:

Figs. 5-8 Please explain what the labels represent in captions. There is no explanation in the paper for "res".

Figures 5-8 have been updated to correct for a mistaken label (see above), and the label "res" in the caption has now been removed due to change in the methods and associated terminology.

Line 176 "Inter-model uncertainty is much higher than intra-model uncertainty". There is no intra-model uncertainty reported in the paper. Please add the missing information.

Intra-model uncertainty describes the uncertainty arising from the internal climatic variability of each ESM. It is represented by the ±1 standard deviation spread in the plots of Figure 1 and Figure S5. Below are reported the revised sentences in which this is explained:

Lines 192-198: "Considering the projections of cumulative carbon sink in Figure 1a and carbon fluxes in Figure S5, the inter-model uncertainty is much higher compared to intra-model uncertainty (which stems from the instrinsic climatic variability expressed in each realization and is represented by the ±1 standard deviation spread in Figure 1 and Figure S5). For the physical variables in Figure 1b-e, intra-model uncertainty is considerably higher than for carbon fluxes and reflects the substantial internal climate variability intrinsic in each simulation. These considerations already highlight that part of the divergence across carbon cycle predictions is related to differences in the land sensitivity to climatic forcings, rather than uncertainties in the evolution of the climate itself. Overall, the climatological variables present a stronger agreement and coherence in the sign of projected changes among ESMs with respect to NBP."

Lines 208-212: "Regarding the intra-model spread, the highest influence of internal climatic variability (±1 standard deviations) is observed in precipitation and shortwave incoming radiation, followed by soil-moisture (spreads in Figure 2c,e and d). This indicates that within the Amazon basin, the major source of uncertainty deriving from internal climate variability is associated to cloud formation and coverage, which is causally associated with the amount of precipitation (thus soil moisture content) and shortwave incoming radiation within the regional domain."

Therefore, what we want to emphasize here is twofold: on the one hand, intra-model uncertainty is higher for climatic variables rather than for carbon fluxes; on the other hand, divergence in carbon cycle projections is mostly related to terrestrial carbon sensitivity to climate, rather than climatic uncertainty itself.

Line 197-200 Please rephrase these sentences.

We have revised the mentioned sentences (lines 226-232 of the revised manuscript) as follows: "This shows that uncertainty in cumulative NBP does not solely stem from uncertainty in single climatic factors; instead, it arises from inconsistencies and differences in how models represent photosynthetic activity, and autotrophic and heterotrophic respiration. Overall, inconsistencies in projected Amazon NBP cannot be simply understood as a consequence of discrepancies in the climatic factors affecting vegetation, as already discussed by Heavens et al., (2013). A point that deserves attention is therefore how the projected carbon sink in the Amazon basin is sensitive to mean-state changes in environmental factors and climatic variability at the interannual timescale."

Reference:

Heavens, Nicholas G., Daniel S. Ward, and M. M. Natalie. "Studying and projecting climate change with earth system models." *Nature Education Knowledge* 4, no. 5 (2013): 4.

Power, S.B. and Delage, F.P., 2018. El Niño–Southern Oscillation and associated climatic conditions around the world during the latter half of the twenty-first century. *Journal of Climate*, *31*(15), pp.6189-6207.

Zhu, Z., Piao, S., Myneni, R.B., Huang, M., Zeng, Z., Canadell, J.G., Ciais, P., Sitch, S., Friedlingstein, P., Arneth, A. and Cao, C., 2016. Greening of the Earth and its drivers. *Nature climate change*, *6*(8), pp.791-795.

---

## Author Response (AR2)

**Reviewer 3:**

In the revision the authors have done a thorough job in responding to the points raised in the previous reviews and have taken new approaches to strengthen their arguments. I truly enjoyed going through the substantially revised manuscript, very straightforward and informative.

We thank the reviewer for the comments on our manuscript, which we have carefully adjusted and improved thanks to the suggestions and the feedbacks from the previous round of revision. We provide here a point-by-point reply to the reviewer's questions and doubts and an updated manuscript with supplementary information.

I was not totally satisfied with the clarity of the method section (Section 2). First, the descriptions of the 1pctCO2-bgc and 1pctCO2-rad experiments could be more clearly presented. These terms first appear in Section 2.1 Table 1, but their explanations are only provided later in Section 2.2 (Lines 135–145), which may confuse readers. I suggest including brief and clear explanations at the point where the terms are first introduced. Meanwhile, the explanations of these terms in Section 2.2 remain unclear. For instance, phrases like "biogeochemically only coupled simulations" (1pct-bgc and ssp585-bgc) and "radiative only coupled simulations" (1pct-rad and ssp585-rad) do not adequately clarify what "1pct," "bgc," and "rad" refer to. I found it necessary to refer back to the original paper (Jones et al., 2016) to fully grasp their meanings.

We thank the reviewer for this comment. We have implemented the following changes in Section 2.1, where the used data are first presented, so that the terminology is now clearer and consistent to what is shown in Table 1, thus making the methodology Section 2.2 more comprehensible without the need to refer to the original paper by Jones et al., 2016:

Lines 89-104:

"[…] .ScenarioMIP simulations (historical and ssp585) aim to reproduce the climatic response to realistic forcing of the historical period and to a prescribed 8.5 $W/m^2$ radiative forcing increase by the end of the 21st century. C4MIP experiments (1pctCO2-bgc, 1pctCO2-rad, ssp585-bgc, ssp585-rad) are idealized concentration-driven carbon-climate simulations generated to better understand and quantify changes in the ocean and land carbon storage and fluxes under different climatic conditions. Specifically, the experiments aim to test the carbon cycle response to the effect of increased $CO_2$ concentrations in the atmosphere (-bgc simulations) and increased radiative forcing resulting from higher atmospheric $CO_2$ concentrations affecting the climate system (-rad simulations). These two categories of simulations differ in their model set-up, so that in the former (-bgc), only the model land and ocean carbon cycle respond to a $CO_2$ increase while the radiation scheme uses a preindustrial $CO_2$ concentration, therefore allowing to test the biogeochemical effect of atmospheric carbon dioxide increase without the associated radiative forcing. Reversely, in -rad simulations the biogeochemical effect is factored out, hence the climate responds to the radiative forcing by increased $CO_2$ concentration, whereas the carbon cycle remains constrained by a preindustrial atmospheric $CO_2$ level. In our analysis, we adopt C4MIP simulations forced either with a 1 % per year increase in atmospheric $CO_2$ concentrations up to four times the preindustrial level of 280 ppm, with no confounding effect of changes in land use, non-$CO_2$ greenhouse gases, and aerosols, or with a standard $CO_2$ pathway from the ssp585 scenario. For the sake of our goal, we count these differences negligible, as considering both 1pct$CO_2$ and ssp585 experiments allows us to have a higher ensemble of data available. […]"

Second, it seems that the terms used in the Methods section and figure captions are inconsistent. For example, long-term sensitivity to climate (γ) include δNBP/δT (γT), δNBP/δmrso (γmrso) and δNBP/δSWin (γSWin) were shown in Section 2.2, 2.3 and 2.4. Confusingly, they seem to become γtas, γmrso and γrsds in Results Section 3.2 (Line 246) without explaining what tas and rsds mean there. In this case, I assume that γtas means γT. However, the caption of Figure 2 refers to γtas as carbon-climate feedbacks to net carbon sink projections, alongside β. This phrasing suggests that $\gamma_{tas}$ may represent the overall carbon–climate feedback. More confusingly, Figure 6 presents $ENSO_{tas}$ and $ENSO_{mrso}$ as parallel variables once again. Please clarify what "tas" exactly means here. If γtas is the overall γ to all climates, please explain how it was derived from the three γT, γmrso and γSWin?

We acknowledge that the terminology may appear confusing to the reviewer, as we did not explicitly describe some acronyms. To be consistent across the different sections of the manuscript, in the revised manuscript we have replaced all the "*tas*" pedix with "*T*" (surface air temperature), as well as "*rsds*" with "*SWin*" (shortwave incoming radiation). We have also clarified that γT represents the overall carbon-climate feedback and is therefore not a derivation of γmrso and γSWin, which have been computed uniquely with the purpose of comparison, providing an equivalent indication of the long-term climate impact on the cumulative carbon sink, as reported in Figure S8:

Lines 266-271:

"The carbon cycle feedback framework aims to describe the positive carbon-climate feedback considering uniquely surface air temperature ($\gamma_T$), which is therefore considered as an overall representation of long-term climate impacts, as shown in Figure 2. We additionally focus on different explainable variables as equivalent terms representing long-term climate impacts ($\gamma_{mrso}$ and $\gamma_{SWin}$). Thus, despite $\gamma_T$, $\gamma_{mrso}$ and $\gamma_{SWin}$ are not to be intended as cumulative long-term impact coefficients, the values of their standardized coefficients are reported in Figure S8 with the purpose of providing a quantitative comparison of the ESMs sensitivity to diverse climatic factors."

On another note, we have rephrased and reviewed the terminology regarding the carbon fluxes variability at interannual timescales associated to ENSO as a function of T and mrso, now reported as $\boldsymbol{\delta NBP_{n34}^T}$ and $\boldsymbol{\delta NBP_{n34}^{mrso}}$. Figure 6, Figure 7 and the whole manuscript now represents these terms in a coherent and clear manner.

While I raise this point for clarity, I don't see it as a major concern, since the main results and conclusions remain compelling and valuable.

Specific comments: In the code availability section, I noticed that the README.md file seemed lack content, and it was a bit challenging to locate the specific code corresponding to each figure. If feasible, it would be appreciated if the authors could consider adding brief guidance or reorganizing the code to enhance clarity for readers.

Thank you for making us notice this. We have restructured the code repository providing guidance in the README.md file and a reorganization of the code expliciting the .ipynb files associated to the generation of the manuscript figures.